# Identification of epigenetic modulators as determinants of nuclear size and shape

Andria C Schibler[1†], Predrag Jevtic[2†], Gianluca Pegoraro[3], Daniel L Levy[2]*, Tom Misteli[1]*

[1]National Cancer Institute, Bethesda, United States; [2]Department of Molecular Biology, University of Wyoming, Laramie, United States; [3]High Throughput Imaging Facility (HiTIF), National Cancer Institute, NIH, Bethesda, United States

*For correspondence:
dlevy1@uwyo.edu (DLL);
mistelit@mail.nih.gov (TM)

[†]These authors contributed
equally to this work

Competing interest: The authors
declare that no competing
interests exist.

Reviewing Editor: Megan C
King, Yale School of Medicine,
United States

**Abstract** The shape and size of the human cell nucleus is highly variable among cell types and tissues. Changes in nuclear morphology are associated with disease, including cancer, as well as with premature and normal aging. Despite the very fundamental nature of nuclear morphology, the cellular factors that determine nuclear shape and size are not well understood. To identify regulators of nuclear architecture in a systematic and unbiased fashion, we performed a high-throughput imaging-based siRNA screen targeting 867 nuclear proteins including chromatin-associated proteins, epigenetic regulators, and nuclear envelope components. Using multiple morphometric parameters, and eliminating cell cycle effectors, we identified a set of novel determinants of nuclear size and shape. Interestingly, most identified factors altered nuclear morphology without affecting the levels of lamin proteins, which are known prominent regulators of nuclear shape. In contrast, a major group of nuclear shape regulators were modifiers of repressive heterochromatin. Biochemical and molecular analysis uncovered a direct physical interaction of histone H3 with lamin A mediated via combinatorial histone modifications. Furthermore, disease-causing lamin A mutations that result in disruption of nuclear shape inhibited lamin A-histone H3 interactions. Oncogenic histone H3.3 mutants defective for H3K27 methylation resulted in nuclear morphology abnormalities. Altogether, our results represent a systematic exploration of cellular factors involved in determining nuclear morphology and they identify the interaction of lamin A with histone H3 as an important contributor to nuclear morphology in human cells.

## Editor's evaluation

In this manuscript the authors describe targeted, imaging-based RNAi screens to identify novel modulators of nuclear size and shape, which are established diagnostic and prognostic indicators of human diseases including cancer. This work provides new insights into the molecules that dictate nuclear morphology tied to chromatin state, the nuclear lamina, and the nuclear envelope. This resource will be broadly valuable to the nuclear cell biology and chromatin biology fields.

## Introduction

In physiological conditions, organelles have predictable morphologies in a cell-type and tissue-specific fashion (*Mukherjee et al., 2016*). In contrast, abnormal and heterogenous organelle morphology is a prominent hallmark of many types of disease (*Bexiga and Simpson, 2013*; *Galloway and Yoon, 2013*). In particular, changes in nuclear size and shape are frequently associated with cancer and aging, and evaluation of aberrant nuclear morphology is routinely used in histology-based diagnostics (*Cantwell and Dey, 2022*; *Mukherjee et al., 2016*; *Pathak et al., 2021*; *Zink et al., 2004*). The comprehensive identification and characterization of cellular factors that determine and maintain nuclear morphology

is important to elucidate the basic molecular mechanisms that determine overall nuclear architecture and to understand how abnormal nuclear morphology contributes to disease.

Nuclear morphology is highly plastic (*Hoskins et al., 2021*; *Versaevel et al., 2012*; *Yoo et al., 2012*). Changes to nuclear shape and size occur during developmental processes such as cellular division, differentiation, and migration (*Skinner and Johnson, 2017*). For example, in early *Drosophila melanogaster* development, embryonic nuclei appear spherical and small while at later stages they assume a more elongated shape with an increase in overall nuclear size. Mutant studies in *Drosophila* identified kugelkern (kuk), a lamin-like nuclear protein, as one factor required for these nuclear morphology changes (*Brandt et al., 2006*). In line with these nuclear shape changes during development, in adult tissues even within the same organism, different cell types often display distinct nuclear shapes and sizes (*Mukherjee et al., 2016*). The cellular factors that determine cell-type and tissue-specific nuclear morphology are only partially understood.

One mechanism implicated in nuclear size control is transport through the nuclear pore complex (NPC) (*Levy and Heald, 2012*). The NPC acts as gateway between the cytoplasm and the nucleoplasm and both negative and positive regulators of nuclear transport have been shown to control the size of the nucleus (*Jevtić et al., 2019*; *Levy and Heald, 2010*; *Levy and Heald, 2012*). In particular, the levels of the nuclear transport factors importin alpha and NTF2 regulate nuclear size in *Xenopus* (*Levy and Heald, 2010*), and inhibition of nuclear exportin XPO1 with leptomycin B has been shown to increase nuclear size (*Kume et al., 2017*; *Neumann and Nurse, 2007*). Furthermore, *Tetrahymena thermophila,* which is a unicellular eukaryote that maintains a macronucleus (MAC) and a micronucleus (MIC), expresses four Nup98 homologs which maintain MAC- and MIC-specific localization. Domain swapping experiments between MAC- and MIC-specific Nup98 proteins resulted in changes in nuclear size indicating that NPC composition can regulate nuclear size (*Iwamoto et al., 2009*). Similarly, the nuclear pore protein ELYS affects nuclear size in human epithelial cells by controlling NPC density and nucleocytoplasmic transport (*Jevtić et al., 2019*).

Obvious candidates that determine nuclear morphology are nuclear lamins and other proteins associated with the nuclear lamina (*Deolal and Mishra, 2021*; *Pathak et al., 2021*). The nuclear lamina is a proteinaceous network of multiple intermediate filament lamin proteins localized at the nuclear periphery between the nucleoplasm and the nuclear envelope. The human genome encodes three lamin genes: *LMNA*, *LMNB1*, and *LMNB2*, of which *LMNA* produces two protein isoforms, lamin A and lamin C (*de Leeuw et al., 2018*; *Karoutas and Akhtar, 2021*). Loss of or mutations in lamina proteins result in dysmorphic nuclei, increased DNA damage, and chromatin organization abnormalities and numerous human diseases, referred to as laminopathies (*Marcelot et al., 2021*; *Shin and Worman, 2022*; *Wong and Stewart, 2020*), which include striated muscle diseases, lipodystrophies, neurological syndromes, and premature aging disorders (*Bonne et al., 1999*; *Kang et al., 2018*; *Karoutas and Akhtar, 2021*; *Méndez-López and Worman, 2012*). One dramatic laminopathy is Hutchinson-Gilford progeria syndrome (HGPS), an exceedingly rare, premature aging disease which results in shortened lifespan, loss of subcutaneous fat, and cardiac abnormalities, among others (*Gordon et al., 2014*). HGPS is caused by a silent point mutation in *LMNA,* that leads to aberrant splicing and to the production of a mutant version of lamin A, referred to as progerin, which carries an internal 50 amino acid deletion at its C-terminus (*Gordon et al., 2014*). Progerin expression in HGPS cells acts in a dominant-negative fashion and results in misshapen nuclei, loss of heterochromatin, and increased endogenous DNA damage (*Gordon et al., 2014*). Furthermore, progerin has also been implicated in normal human aging (*Scaffidi and Misteli, 2006*).

In addition to the lamin proteins, chromatin has also been implicated in nuclear morphology. Early observations showed that in *T. thermophila*, a specific nuclear histone linker protein is required for the reduced nuclear size in micronuclei (*Allis et al., 1979*; *Shen et al., 1995*). In addition, epigenetic readers and modifiers affect nuclear size. For example, overexpression of the histone H3 acetyltransferase BRD4 increases nuclear size in HeLa cells (*Devaiah et al., 2016*). In MCF10A breast epithelial cells, a number of epigenetic and chromatin factors, including several core histones, have been shown to affect nuclear morphology (*Tamashunas et al., 2020*). Furthermore, recent observations suggest that the interaction of chromatin with lamins contributes to determining nuclear morphology (*Karoutas et al., 2019*; *Stephens et al., 2019a*). Single cell micromanipulation revealed two independent responses to mechanical forces, nuclei responded to small manipulations through chromatin and larger manipulations through lamin A/C (*Stephens et al., 2017*). Further studies of the

role of chromatin in regulating nuclear morphology found chromatin to regulate nuclear dynamics and rigidity. In particular, manipulating the relative levels of euchromatin and heterochromatin altered nuclear architecture (*Stephens et al., 2019a*; *Stephens et al., 2018*). Furthermore, in cells with perturbed chromatin or lamins, increased heterochromatin suppressed nuclear blebbing and maintained nuclear rigidity (*Stephens et al., 2019a*; *Stephens et al., 2019b*). Close interplay between lamins and chromatin is also illustrated by the observation that loss of the lysine acetyltransferase MOF or its associated NSL-complex members KANSL2 or KANSL3 leads to altered mechanical properties of nuclei (*Karoutas et al., 2019*). While this effect appears to be due to reduced lamin acetylation, the observed changes in nuclear morphology are accompanied by alterations of the epigenetic chromatin landscape (*Karoutas et al., 2019*). These observations strongly suggest that nuclear size and shape are not determined by a single factor but rather are the result of an intricate interplay of architectural nuclear proteins with chromatin.

Nuclear morphology changes are routinely observed in pre-neoplastic and malignant cancer tissues. For example, tumor cells commonly display nuclear morphology abnormalities compared to nuclei in surrounding tissue (*Chow et al., 2012*). Anecdotal observations have identified multiple factors implicated in nuclear morphology and misregulation, and some of these factors have been linked to oncogenesis. For instance, mutations or alterations in expression of components of the LINC complex (linker of nucleoskeleton and cytoskeleton), such as SYNE1 and SYNE2, which localize to the nuclear envelope and connect to the nuclear lamina, result in misshapen nuclei (*Lüke et al., 2008*; *Zhang et al., 2007*). Alterations to SYNE1 and SYNE2 have been observed in colorectal (*Yu et al., 2015*), lung (*Ahn et al., 2014*), breast (*Zuo et al., 2011*), glioblastoma (*Masica and Karchin, 2011*), and ovarian cancer (*Doherty et al., 2010*). In addition, misexpression or mislocalization of nuclear lamins has been documented in both cancerous cells and tissues (*Broers et al., 1993*; *Moss et al., 1999*). Furthermore, lamin A/C is overexpressed in colorectal cancers, where it correlates with poor prognosis, and overexpressing GFP-lamin A in colorectal cancer cells increases cell motility (*Willis et al., 2008*). In contrast, reduced lamin A/C expression is documented in carcinomas of the esophagus, as well as in breast, cervical, and ovarian cancers (*Prokocimer et al., 2006*).

Despite the fundamental nature of nuclear morphology and its link to disease, our knowledge of nuclear components and mechanisms that regulate nuclear size and shape is very limited (*Cantwell and Dey, 2022*; *Mukherjee et al., 2016*). Here, we have used an imaging-based high-throughput RNAi screen to systematically identify epigenetic- and nuclear envelope-associated factors that affect nuclear morphology in multiple cell lines. We find known and novel determinants of cell type-specific and general nuclear morphology and uncover a novel mechanism of lamin-chromatin interactions mediated via histone H3 and its epigenetic modifications as a critical modulator of nuclear morphology.

## Results
### An imaging-based screen to identify determinants of nuclear size and shape

We developed an imaging-based RNAi screen to identify and characterize novel cellular factors that regulate nuclear morphology, particularly nuclear shape and size (*Figure 1A*). We focused on nuclear proteins that may affect nuclear morphology via interactions that take place either at the nuclear membrane or within the nucleus. To this end, we screened karyotypically normal hTERT-immortalized dermal fibroblasts against two siRNA oligos libraries targeting 346 proteins that localize to the nuclear membrane and 521 proteins involved in epigenetic and chromatin regulation, respectively (see Materials and methods; *Supplementary file 1A*). For each gene targeted in the library, immortalized fibroblasts were reverse transfected in 384-well format with three unique siRNAs in three separate wells. Seventy-two hr after siRNA reverse transfection, cells were fixed, permeabilized, and immuno-stained with antibodies against lamin A/C and lamin B1, and with DAPI to visualize DNA and to assess changes to nuclear morphology (see Materials and methods) (*Figure 1A*). Images were acquired on a high-throughput spinning disk confocal microscope and nuclear morphology was quantified using an automated image analysis pipeline (see Materials and methods) (*Figure 1A*). To assess multiple aspects of nuclear morphology, nuclear length, width, area, and circularity were measured simultaneously for all samples, treated as independent parameters, and used for hit identification (*Figure 1A*, *Supplementary file 1D*). The screen was performed in biological duplicates and results were strongly correlated

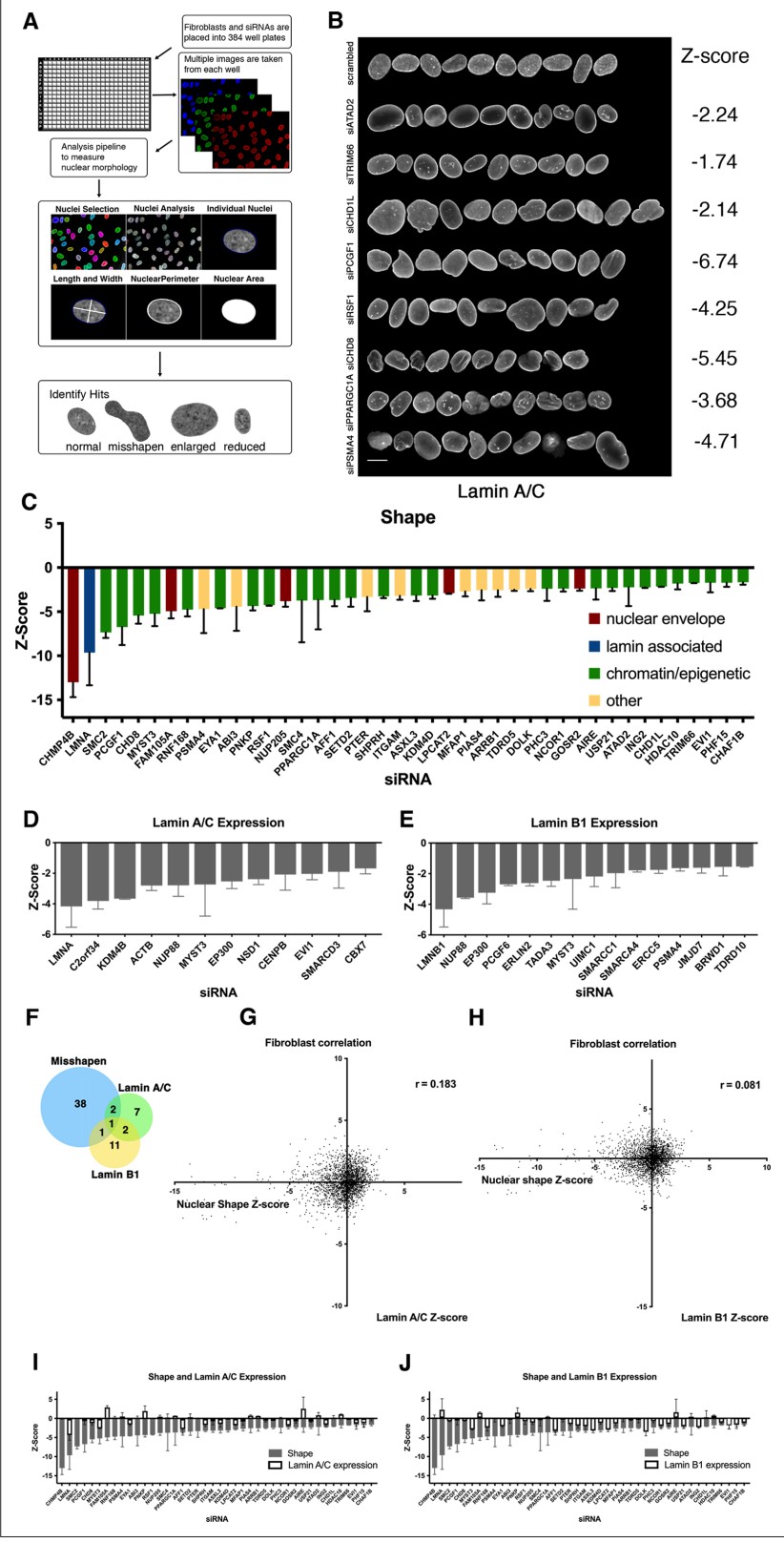

**Figure 1.** A high-throughput image-based screen for nuclear size and shape determinants. (**A**) Schematic overview of the nuclear morphology screen to identify genes required for proper nuclear shape and size. Cells were cultured in the presence of siRNAs targeting specific genes in 384-well plates, fixed, and stained with specific antibodies to visualize lamin A/C, lamin B1, or DAPI. Cells were imaged using high-throughput microscopy and an image

*Figure 1 continued on next page*

*Figure 1 continued*

analysis pipeline was developed to segment individual nuclei, measure nuclear morphology, and lamin A/C and lamin B1 intensity. Datasets were analyzed to identify misshapen, enlarged, and shrunken nuclei. Nuclear morphology changes were measured as Z-scores. Typically, more than 250 nuclei were analyzed per sample. The fibroblast screen was performed in two biological replicates on different days. (**B**) Representation of normal nuclei and nuclear shape hits identified by high-throughput screening. Nuclear shape abnormalities are visualized by lamin A/C antibody staining. Scale bar: 10 μm. (**C**) Nuclear shape hits were calculated by scoring circularity (circularity = $4\pi$Area/perimeter$^2$). Z-scores were generated to compare hits across the screen. Nuclear shape hits were identified by a median Z-score of −1.5 or less. At least 250 nuclei were analyzed per sample. Error bars indicate the standard deviation. (**D**) Nuclear intensity of lamin A/C was assessed by calculating Z-scores of changes in lamin A/C expression on a per well basis. Lamin A/C hits were identified by median Z-scores of −1.5 or less. Error bars indicate the standard deviation. (**E**) Nuclear intensity of lamin B1 was assessed by calculating median Z-scores of changes in lamin B1 expression of −1.5 or less per well. Error bars indicate the standard deviation. (**F**) A comparison between lamin A/C and lamin B 1 expression hits and nuclear shape hits indicates lamin levels are not affected in most of nuclear shape hits. Error bars indicate the standard deviation. (**G**) Correlation between nuclear shape and lamin A/C expression of the Z-score of each parameter. Nuclear shape values relative to lamin A/C expression show little to no correlation. Spearman's coefficient (r=0.183). (**H**) Scatterplot of nuclear shape values relative to lamin B1 expression shows little to no correlation in lamin B1 protein expression levels and nuclear shape scores in fibroblasts. Spearman's coefficient (r=0.081). (**I**) Nuclear shape hits in panel C with gray bars indicating the Z-score for circularity. Lamin A/C Z-score values are overlayed with white bars indicating lamin A expression changes. Error bars indicate the standard deviation. (**J**) Nuclear shape hits in panel C displaying Z-score values. Lamin B1 Z-score values are overlayed with white bars to indicating levels of lamin B1. Error bars indicate the standard deviation.

The online version of this article includes the following figure supplement(s) for figure 1:

**Figure supplement 1.** Reproducibility of nuclear morphology screen using human fibroblast cells.

**Figure supplement 2.** A schematic overview of nuclear shape and nuclear size image analysis pipeline and generation of Z-scores.

**Figure supplement 3.** Nuclear shape hits that affected cell number.

**Figure supplement 4.** Nuclear shape Z-scores in comparison to nuclear shape, area, and perimeter raw scores of fibroblast hits.

**Figure supplement 5.** Single cell values of nuclear shape hits compared to control cells.

(*Figure 1—figure supplement 1*). A non-targeting siRNAs was used as a negative control, and a mix of siRNA oligos with a lethal phenotype was used as a positive technical control to measure transfection efficiency (see Materials and Methods). In addition, an siRNA targeting *LMNA* was used as a positive control based on the previously demonstrated role *LMNA* plays in maintaining nuclear shape (*Lammerding et al., 2004*) and as a control for fluorescence intensity levels of immuno-stained lamin A/C. Single cell measurements were averaged on a per well basis. Mean per well values were normalized on a per plate basis using the B-score formula. B-score values were then standardized across a single replicate as robust Z-scores. Z-scores from the two replicates were averaged to obtain a mean Z-score per siRNA oligo. Z-scores measure the number of standard deviations a sample was away from the mean of all the samples in the library (*Figure 1—figure supplement 2*). Genes were defined as hits if their median Z-score (two out of three siRNA oligos targeting the same gene) was ±1.5 (see Materials and Methods for details). To eliminate false-positive hits due to cell death, altered cell cycle behavior, or indirect proliferation effects, we excluded any hits with a cell number Z-score of less than −2. As expected, excluded hits included spindle assembly checkpoint components *MAD2L1*, *BUB1*, and cell cycle-related kinases *AURKA* and *AURKB*, among others (*Figure 1—figure supplement 3*).

## Nuclear shape determinants

The phenotypic effect of siRNA knockdowns on nuclear morphology in the screen was quantified by Z-score analysis of the nucleus roundness parameter (roundness = $4\pi$Area/perimeter$^2$, *Figure 1B and C*; see Materials and Methods). Using this parameter, and excluding cytotoxic or cytostatic genes (see above), we identified 42 genes as positive hits in the primary screen, corresponding to a hit rate of 4.8% (42/867, *Figure 1C*, *Figure 1—figure supplement 4*). Nuclear shape regulators identified in the primary screen included genes whose products localize to the nuclear membrane or nuclear lamina such as *CHMP4B*, *LMNA*, or the nuclear pore component *NUP205*. Interestingly, 64% (27/42) of the

effectors altering nuclear shape were genes whose products are involved in epigenetic modifications such as the polycomb component *PCGF1*, the histone acetyltransferase *MYST3*, the histone methyltransferase *SETD2*, or the ring finger protein *RNF168* (*Figure 1C*). Single cell analysis indicated that changes in Z-scores were the result of population-wide changes in the circularity parameter measured, rather than alterations in a subpopulation of cells (*Figure 1—figure supplement 5*).

Loss of lamins and lamin mutations have been linked to nuclear morphology changes in previous studies (*De Sandre-Giovannoli et al., 2003*; *Eriksson et al., 2003*; *Lammerding et al., 2004*). As expected, and reassuringly, *LMNA* was identified in the siRNA oligo library as the second strongest hit in our screen (*Supplementary file 2A*). To more broadly test how changes in lamin levels relate to alterations in nuclear shape, we mined our screening data for factors that lowered lamin A/C and lamin B1 levels (*Figure 1D–F*). Twelve siRNA targets reduced lamin A/C level by a Z-score of at least 1.5 (*Supplementary file 2B*) and 15 targets reduced lamin B levels by a Z-score of 1.5 or more (*Supplementary file 2C*). Factors affecting both lamin A/C and lamin B1 levels include the nuclear pore protein *NUP88*, histone acetyltransferase *MYST3*, and the transcription co-factor and histone acetyltransferase *EP300*. To test whether the identified shape effectors exerted their effect on nuclear shape primarily via altering lamin levels, we assessed lamin A/C or lamin B1 in all shape effectors using quantitative imaging (*Figure 1I and J*). Remarkably, of the 42 shape hits, only 4 affected lamin A/C and lamin B1 expression (*Figure 1F–J*). This groups included the histone acetyltransferase *MYST3* that affects nuclear shape, lamin A/C, and lamin B1 levels, as well as the zinc-finger transcription factors *EVI1* and *LMNA* which affect both nuclear shape and lamin A/C levels, and the proteasome component *PSMA4* which affects nuclear shape and lamin B1 levels. However, for the majority of shape effectors (38/42), lamin A/C and lamin B1 levels were unaffected indicating that the majority of shape hits did not exert their effect on shape indirectly via altering lamin levels (*Figure 1F–J*). These results identify known modulators of nuclear shape, including factors which are accompanied by reduction of lamins, but more importantly, they point to a larger set of novel lamin-independent nuclear shape factors, including numerous epigenetic modifiers.

## Nuclear size determinants

In a complementary approach, we identified cellular factors that determined nuclear size. While nuclear size and shape are related features of nuclear morphology (*Figure 2A*), we hypothesized that cellular factors exist that independently affect size or shape. While the nuclear cross-sectional area has previously been experimentally shown to be a good proxy for nuclear size in many systems (*Edens and Levy, 2014*; *Jevtić and Levy, 2015*; *Levy and Heald, 2010*; *Mukherjee et al., 2020*; *Vuković et al., 2016*), our imaging approach does not include information about nuclear height or volume, and thus uses the cross-sectional nuclear area as a proxy measurement for nuclear size (*Figure 2A*). siRNA knockdown of 50 of the 867 genes affected nuclear size using a median Z-score threshold of ±1.5, corresponding to a hit rate of 5.7% (*Figure 2B and C*). Among the size effectors, knockdown of 36 of these factors increased nuclear size whereas 14 resulted in smaller nuclei (*Figure 2B and C*). As observed for shape effectors, a large number of hits (52%; 26/50) were chromatin or epigenetic modifiers. Factors which increased nuclear size included genes encoding nuclear pore proteins such as *NUP205*, *NUP62*, *NUPL1*, and *NUP85* as well genes that encode components of histone modifiers such as *SUPT7L* and *PRMT2*. Factors that decreased nuclear size included the PhD-finger protein *PHF10*, the deacetylase *SIRT4*, the acetylation reader *BRD2*, and the histone methyltransferase *MLLT10* (*Figure 2B*). Much like for nuclear shape, nuclear size determinates did not exert their effect via lamins, because of the 50 size determinants, none altered lamin levels (*Figure 2D*).

Given the relationship of nuclear size and shape, we cross-compared factors that affected both size and shape (*Figure 2D*). Remarkably, we find almost no overlap between nuclear size determinants and nuclear shape effectors. Of the 42 shape effectors and 50 total size effectors, only one, the nuclear pore component *NUP205*, overlapped (*Figure 2D*). These results suggest that nuclear shape and size are regulated by separate cellular pathways.

We extended our analysis to a second cell type. In an identical screen using MCF10AT breast cancer cells (*Figure 3*), we identified 34 factors needed for proper nuclear shape in MCF10AT cells (*Figure 3A and B*; *Supplementary file 3D*). Interestingly, only three, *LMNA*, and the transcription factors *EYA1* and *TRIM6*, overlapped with the shape determinants identified in fibroblasts (*Figure 3A and B*). More generally, among all shape effectors in both screens, nuclear shape scores showed

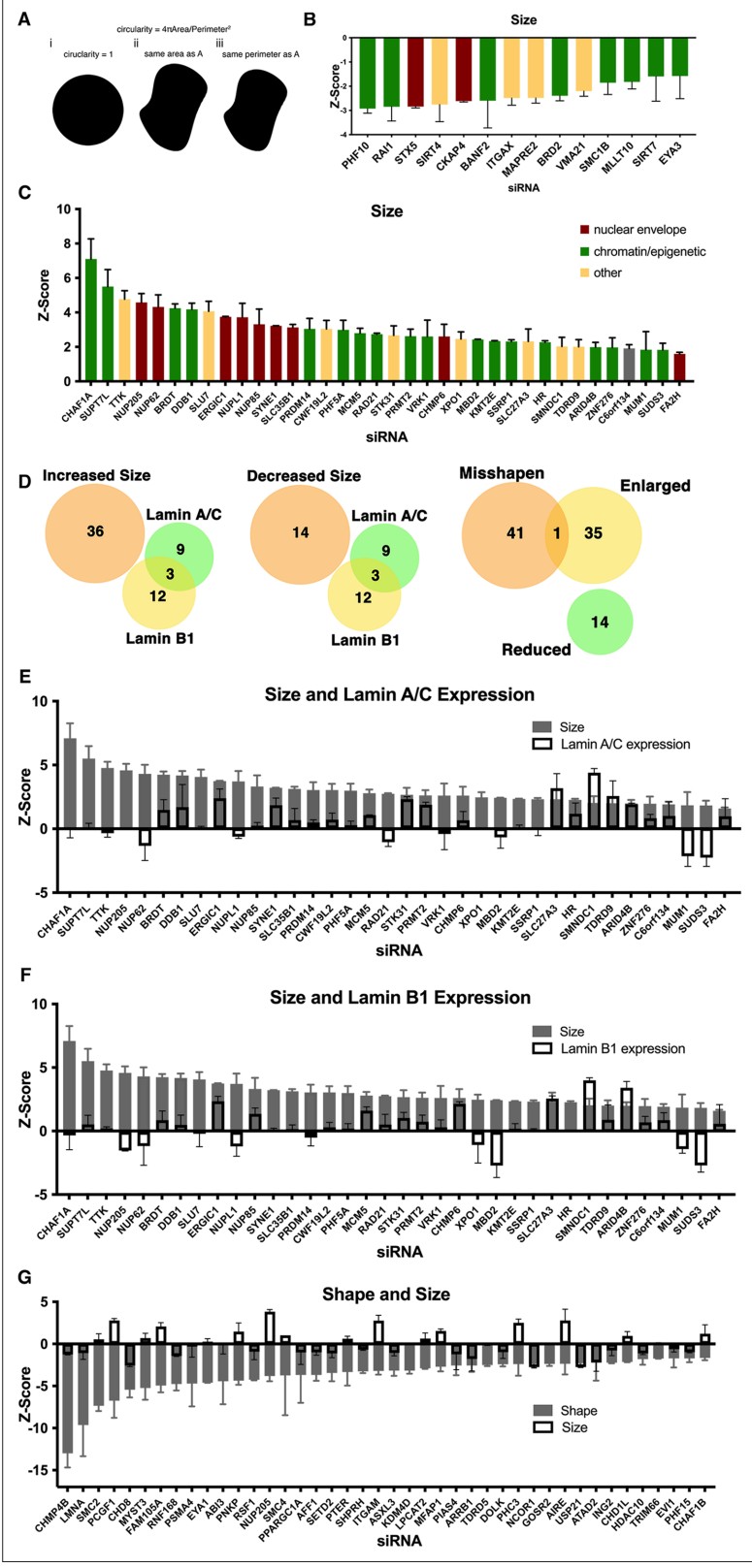

**Figure 2.** Identification of nuclear size determinants in human fibroblast cells. (**A**) A diagram of the relationship between circularity used to calculate nuclear shape and nuclear perimeter and area revealing the possibility that nuclear size might affect nuclear shape. Nuclear size hits were calculated by (circularity = 4πArea/perimeter²). Panel i shows a perfect circle with a circularity score of 1. Panel ii shows the same area as in panel A but an increased

*Figure 2 continued on next page*

*Figure 2 continued*

perimeter. Panel iii shows the same perimeter as panel A but a decreased area. If perimeter is a cellular constant, shape hits would correlate with a decrease in nuclear size. (**B**) Z-scores of hits displaying a decrease in nuclear size. Nuclear size hits maintain a median Z-score of –1.5 or less. At least 250 nuclei were analyzed per sample. Error bars indicate the standard deviation. (**C**) Z-scores of hits displaying an increase in nuclear size. Nuclear size hits maintain a median Z-score of 1.5 or greater. At least 250 nuclei were analyzed per sample. Error bars indicate the standard deviation. (**D**) Relationship of nuclear size hits and lamin A/C and lamin B1 hits. Nuclear shape hits show little overlap with nuclear size hits. (**E**) Nuclear size hits in panel C with gray bars indicating the Z-score for hits displaying an increase in size. Lamin A/C Z-score values overlayed with white bars indicating lamin A/C level. Error bars indicate the standard deviation. (**F**) Bar graph of nuclear size hits displaying Z-score values. Lamin B1 Z-score values are overlayed with white bars indicating levels of lamin B1. Error bars indicate the standard deviation. (**G**) Z-score values of nuclear shape hits in gray. Nuclear size Z-score values are overlayed with white bars to show nuclear size varies among nuclear shape hits. Error bars indicate the standard deviation.

The online version of this article includes the following figure supplement(s) for figure 2:

**Figure supplement 1.** Nuclear morphology data using human fibroblast cells reveal lack of correlation between lamin expression and nuclear morphology features.

limited correlation (r=0.301) (***Figure 3D***). Cell-type differences in nuclear shape effectors were confirmed by direct side-by-side comparison of hits in fibroblast and MCF10AT cell lines (***Figure 3—figure supplement 1***). Several hits displayed nuclear shape morphology changes in one cell line but not the other. This observation was prominent for KAT2B (PCAF) which is expressed in both fibroblasts and MCF10AT, but after its knockdown only MCF10AT cells displayed misshapen nuclei (***Figure 3—figure supplements 1 and 2***). The same was true for RNF168, which is expressed in both cell lines but only fibroblasts displayed strong misshapen nuclear morphology upon knockdown (***Figure 3—figure supplement 1G***). These observations are in line with previous results from an siRNA screen in the human breast epithelial cell line MCF10A (***Imbalzano et al., 2013***), which revealed a number of hits affecting nuclear shape (***Tamashunas et al., 2020***). Interestingly, most hits do not overlap with our findings suggesting that these screens are not saturated or that nuclear shape change detection methods monitor distinct nuclear features.

Similarly, comparison of nuclear size effectors between fibroblasts and MCF10AT cells also showed little overlap (r=0.4228) (***Figure 3F***). Of the fibroblast and MCF10AT nuclear size hits, only four overlapped (the chromatin assembly factor *CHAF1A*, DNA binding protein *DDB1*, transcription factor *PRDM14*, and transport protein *SLC27A3*) (***Figure 3—figure supplement 3***). While *SLC27A3* functions at the nuclear membrane, *CHAF1A, DDB1,* and *PRDM14* act through chromatin. Along with the cell type-specific effectors of nuclear size and shape, limited correlation between the effectors of lamin A/C and lamin B levels was found between cell types (***Figure 3C and D***).

To further analyze nuclear shape and size effectors between cell lines, we employed STRING, a protein functional association network and pathway tool (***Szklarczyk et al., 2023***). In fibroblasts, shape hits represented a highly connected node of core components of condensin (SMC2, SMC4) and polycomb repressive complex members PCGF1 and PHC3. Effectors of nuclear shape in MCF10AT cells formed highly connected regions representing nucleoporins NUP155 and NUP93 (***Figure 3—figure supplement 4***). When comparing nuclear size hits in fibroblast cells, nucleoporins NUP205, NUP62, NUP85, and NUPL1 showed increased connectivity, while DNA replication components RPA3 and PCNA were associated in MCF10AT with nuclear size hits (***Figure 3—figure supplement 5***). Taken together these results point to cell-type specificity in effectors of nuclear morphology. To confirm knockdown efficiency and cell-type specificity, we tested knockdown efficiency of a subset of hits. Typical knockdown efficiencies were in the 60–90% range in fibroblasts and MCF10AT cells (***Figure 3—figure supplement 2***). To rule out possible siRNA off-target effects, hits from the primary screen were further validated in a secondary screen using siRNAs against the same genes, but with different target sequences and oligo chemistry (***Supplementary file 3A, B and C***).

## Lamin A directly interacts with histone H3

Given our identification of numerous epigenetic modulators as determinants of nuclear size and shape, combined with the lack of accompanying changes in lamins, we considered that chromatin-lamin interactions might mediate the observed size and shape effects. In particular, based on the

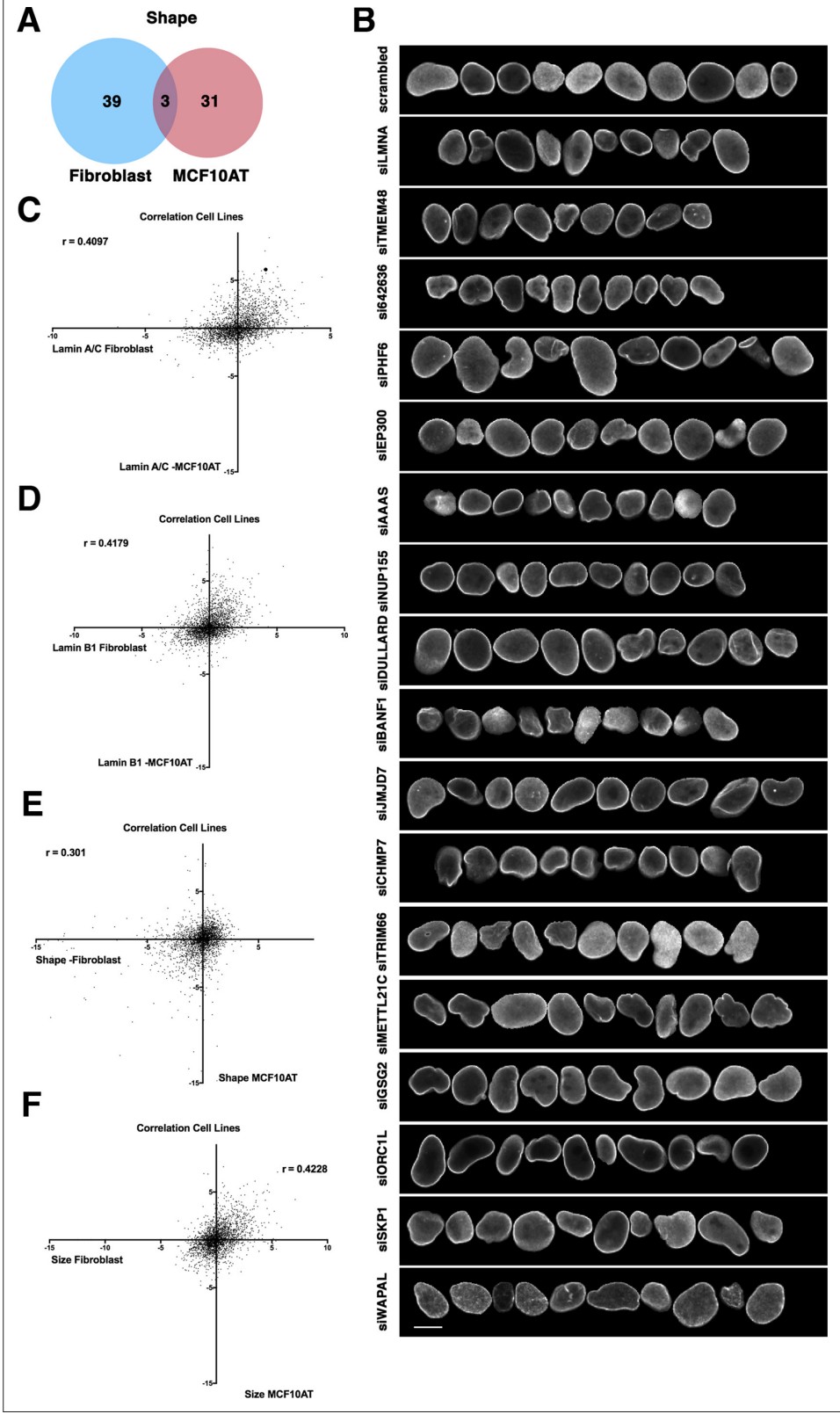

**Figure 3.** Cell-type specificity of effector hits. (**A**) Little overlap of hits for nuclear shape changes in immortalized human fibroblast cells compared to nuclear shape hits for the breast epithelial cell line MCF10AT. The MCF10AT screen was performed in two biological replicates on different days. (**B**) Representative normal nuclei and nuclear shape hits of MCF10AT cells identified by high-throughput nuclear morphology screen and analysis. Nuclear shape

*Figure 3 continued on next page*

*Figure 3 continued*

abnormalities are visualized by lamin A/C antibody staining. Scale bar = 10 µm. (**C**) Lamin A/C Z-score values in fibroblasts compared with lamin A/C Z-score values in MCF10AT cells using the same siRNA target reveals little correlation between values. Spearman's coefficient (r=0.4097). (**D**) Lamin B1 Z-score values in fibroblasts compared with lamin B1 Z-score values in MCF10AT cells using the same siRNA target reveals little correlation between values. Spearman's coefficient (r=0.4179). (**E**) Nuclear shape Z-scores of fibroblast and MCF10AT cell lines reveal a lack of overlap between the same siRNA targets. Spearman's coefficient (r=0.301). (**F**) Nuclear size Z-score values in fibroblast cells compared with nuclear size Z-score values in MCF10AT cells using the same siRNA target reveals little correlation between data points. Spearman's coefficient (r=0.4228).

The online version of this article includes the following figure supplement(s) for figure 3:

**Figure supplement 1.** Validation of nuclear shape hits in fibroblast and MCF10AT cell lines.

**Figure supplement 2.** Validation of knockdown efficiency in both fibroblast and MCF10AT cell lines.

**Figure supplement 3.** Identification of nuclear size determinants in MCF10AT cells.

**Figure supplement 4.** Functional protein association analysis using STRING for nuclear shape hits (**A**) in human fibroblast cells, (**B**) in MCF10AT cells.

**Figure supplement 5.** Functional protein association analysis using STRING for nuclear size hits (**A**) in human fibroblast cells, (**B**) in MCF10AT cells.

**Figure supplement 6.** Correlation plots of nuclear morphology data using MCF10AT human breast epithelial cells.

**Figure supplement 7.** Identification of nuclear shape determinants in MCF10AT cells.

---

identification in our primary screens of several post-translational modifiers of histone H3, including the histone H3K36-specific lysine methyltransferase *SETD2*, the histone H3K9 lysine demethylase *KDM4D*, and the histone deacetylase *HDAC10*, we hypothesized that histone H3-lamin interactions may contribute to maintaining nuclear size and shape.

To test this hypothesis, we first asked whether lamin A/C could bind directly to chromatin in vitro. Lamin A and lamin C consist of an N-terminal head domain followed by a long rod-like domain in the central region of the proteins and prior observations showed that the C-terminus of lamin A maintains an IgG-like fold domain and directly binds to DNA (*Stierlé et al., 2003*). For that reason, and because many disease mutations affecting nuclear morphology localize to this region (*Dittmer and Misteli, 2011*; *McKenna et al., 2013*), we probed for a direct physical interaction of the C-terminal region of lamin A/C with chromatin (*Figure 4*). We generated GST fusions proteins of various fragments of lamin A or C, purified them, and incubated them with histones derived from calf thymus to test for direct binding to histones (*Figure 4A and B*). We find that GST-lamin A containing the entire Ig-fold and C-terminus (aa 389–646) binds directly to histone H3, but not to the other core histones (*Figure 4B*). GST-lamin A lacking the C-terminal 80aa (aa 389–566) or the C-terminal region of GST-lamin C (aa 389–572) did not bind to histone H3, suggesting that the unique portion of lamin A present in the C-terminus tail is required for the histone H3 interaction (*Figure 4B*). However, while the lamin A tail was required for H3 binding, this region (aa 565–646) was not sufficient for binding (*Figure 4B*), possibly due to its disordered nature (*Qin et al., 2011*). We conclude that lamin A can directly interact with core histone H3 via its C-terminal tail along with a portion of the homologous region present in both lamin A and lamin C.

To more precisely identify the region of the lamin A C-terminus required for the interaction with histone H3, multiple truncation mutants were generated and used in in vitro histone binding assays. GST-lamin A (aa 389–638) bound as well to histone H3 as the full-length lamin A tail (GST-lamin A, aa 389–646), while the aa 389–626 region did not (*Figure 4C*). These experiments identify aa 627–638 as required for the lamin A-histone H3 interaction. Interestingly, this is the region deleted in the lamin A mutant isoform that causes the premature aging disorder HGPS, which is characterized by extensive nuclear shape aberrations, including prominent nuclear lobulations and altered H3K9 and H3K27 methylation (*Goldman et al., 2004*; *Shumaker et al., 2006*). In line with a possible role of lamin A-histone H3 interactions in HGPS, GST-progerin (aa 506–664 D50) (*Figure 4—figure supplement 1A*) did not bind H3 (*Figure 4—figure supplement 1B*).

Binding of lamin A to naked DNA had previously been shown to occur in the context of lamin A dimers (*Stierlé et al., 2003*). To ask whether dimerization is required for lamin A-H3 interactions, we incubated GST-lamin A fusion proteins with the reducing agent DTT to inhibit dimerization. The

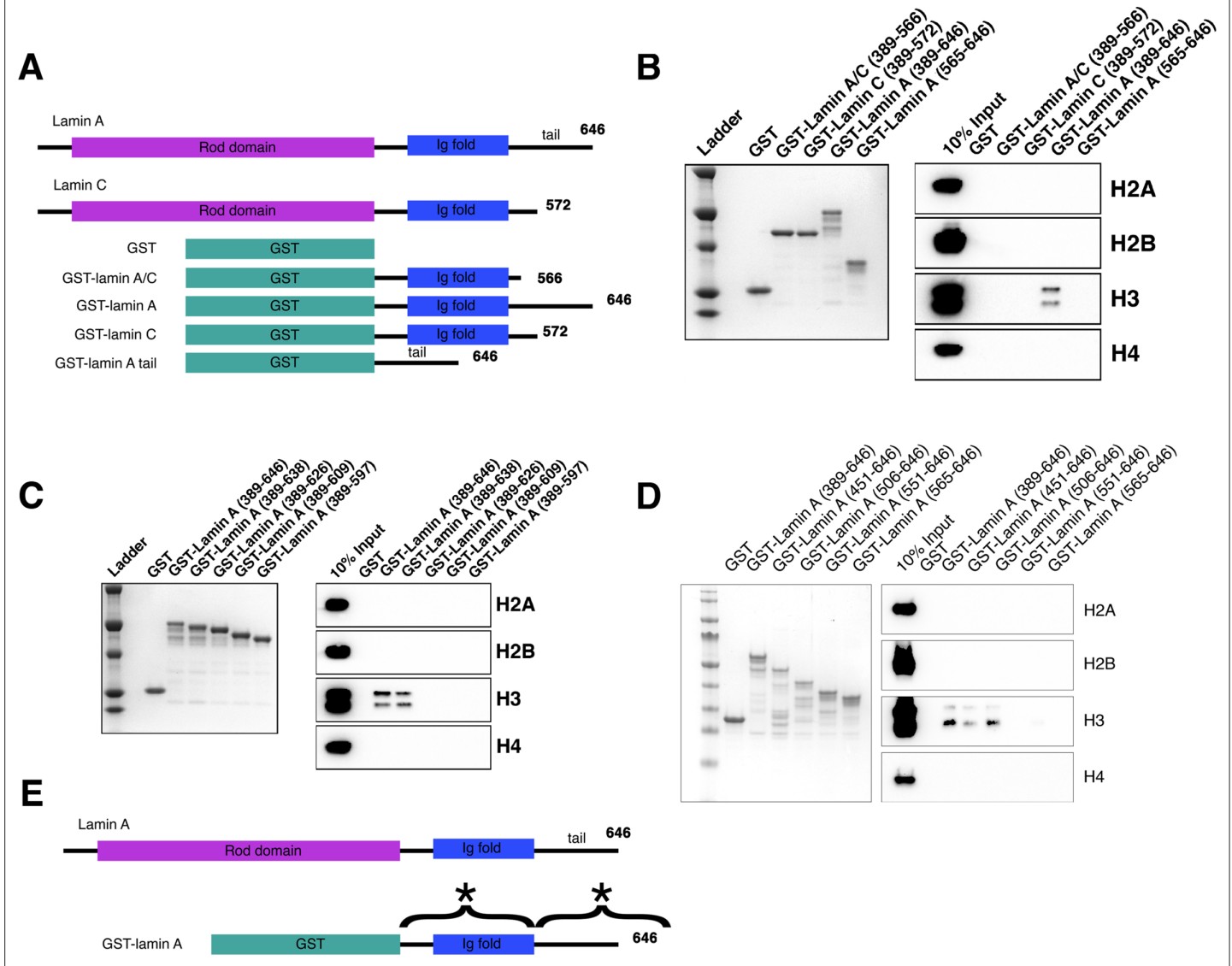

**Figure 4.** In vitro binding of lamin A to histone H3. (**A**) A diagram of constructs used in binding assays. All histone pulldown assays were performed at least three times. (**B**) Colloidal staining of purified recombinant proteins and histone pulldown assays. GST-lamin A (389–646) directly binds to histone H3 but not histones H2A, H2B, and H4. The portion of lamin A and lamin C that is homologous (GST-lamin A/C [389–566], GST-lamin C [389–572], and the GST-lamin A truncation containing the C-terminal tail [GST-lamin A 565–646]) did not bind histones. (**C**) Colloidal staining of purified recombinant proteins and histone pulldown assay. The C-terminal portion of lamin A is required for binding histone H3. Full-length GST-lamin A (389–646) and the truncated GST-lamin A (389–638) bound histone H3 but not histones H2A, H2B, and H4. Further truncations to the lamin A tail (389–626), (389–609), and (389–597) did not interact with histones identifying the portion of the C-terminal tail essential for binding histone H3 as aa 638–646. (**D**) Colloidal staining of purified recombinant proteins and histone pulldown assay. The N-terminal portion of lamin A required for binding histone H3. Full-length GST-lamin A (389–646) and the truncated GST-lamin A (451–646) and GST-lamin A (506–646) bound histone H3 but not histones H2A, H2B, and H4. Further truncations to the GST-lamin A (551–646) and GST-lamin A (565–646) did not interact with histones identifying the portion of the N-terminus essential for binding histone H3 as aa 506–550. (**E**) A schematic summary of the two regions required for lamin A-H3 interactions marked with (*).

The online version of this article includes the following source data and figure supplement(s) for figure 4:

**Source data 1.** Source data for *Figure 4B*.

**Source data 2.** Source data for *Figure 4C*.

**Source data 3.** Source data for *Figure 4D*.

**Figure supplement 1.** Lamin C inhibits lamin A-H3 interactions.

**Figure supplement 1—source data 1.** Source data for *Figure 4—figure supplement 1B*.

**Figure supplement 1—source data 2.** Source data for *Figure 4—figure supplement 1C*.

*Figure 4 continued on next page*

*Figure 4 continued*

**Figure supplement 1—source data 3.** Source data for *Figure 4—figure supplement 1D*.

**Figure supplement 1—source data 4.** Source data for *Figure 4—figure supplement 1E*.

addition of DTT increased lamin A-H3 interactions suggesting lamin A binds to histone H3 as a monomer and dimerization limits its binding (*Figure 4—figure supplement 1C*). Furthermore, since both lamin A and lamin C can form dimers, bind to DNA, are co-expressed in vivo, and lamin C does not bind histone H3, we asked whether lamin C binding to lamin A can inhibit histone H3 binding via dimer formation. Co-incubation of the C-termini of GST-lamin A (aa 389–646) and GST-lamin C (aa 389–572) in a histone pulldown assay reduced lamin A-histone H3 binding (*Figure 4—figure supplement 1D*). The addition of DTT reversed inhibition, suggesting lamin C inhibits lamin A through dimerization at cysteine residues (*Figure 4—figure supplement 1D*). The inhibitory effect was due to lamin A-lamin C interactions, rather than due to the presence of the GST moiety, since mutation of C522A in lamin A alleviated the inhibitory effect of lamin C (*Figure 4—figure supplement 1E*). These observations demonstrate direct interaction of lamin A with core histone H3.

## Interaction of lamin A with histone H3 is sensitive to epigenetic modifications

To further identify how lamin A binds to histone H3, we utilized a histone peptide binding array to probe the effect of histone tail modifications on lamin A-histone interactions (*Figure 5*). The array consists of 384 unique histone peptides spotted onto a slide representing peptides of histones H2A, H2B, H3, and H4 with multiple common modifications including serine/threonine phosphorylation, lysine acetylation, and methylation among others (see Materials and Methods). Binding assays using recombinant lamin A (aa 506–646) added to the array confirmed interaction of lamin A with histone H3, and also revealed preferential binding to several histone modifications, including combinations of modifications (*Figure 5A*; *Supplementary file 3E*). In fact, the peptides that showed the strongest lamin A signal contained a combination of modifications (*Supplementary file 3E*). Specifically, lamin A bound preferentially to peptides which contained a methyl-methyl modification signature, including histone H3R8me2/K9me2 (*Figure 5B*), H3K26me2/K27me2 (*Figure 5C*), and histone H4R19me2/K20me1 (*Figure 5D*). In line with this observation, GST-lamin A binding to peptides containing only a single modification, such as methylated arginine or methylated lysine alone, was reduced when compared to the three dually modified peptides (*Figure 5B–D*; *Supplementary file 3E*). For example, the GST-laminA signal was ~5-fold greater for histone H3R8me2/K9me2 than for histone H3K9me2. While most peptides with the strongest GST-lamin A signal contained dual methyl marks, we did find lamin A also bound to acetylated histone H3 and H4 peptides (*Supplementary file 3E*). These included histone H4K12ac/K16ac, H3K27ac, and H3K4ac peptides although at reduced levels (*Supplementary file 3E*). These data indicate that methyl-methyl motifs represent a target sites for lamin A-chromatin binding.

## Histone H3.3 mutants result in nuclear shape and size abnormalities

The binding of lamin A to histone H3 is of interest since mutations in this histone variant have been implicated in disease. Mutations to histone H3.3 were first found in pediatric high-grade glioma and later in chondrosarcomas and giant cell tumors of the bone (*Weinberg et al., 2017*). Recently, mutations to histone H3.3 have been associated with congenital disorders such as craniofacial and brain abnormalities, and developmental delay among others (*Bryant et al., 2020*). Interestingly, most of these diseases, including gliomas and chondrosarcomas, are characterized by changes in nuclear morphology (*Nafe et al., 2003*; *Welkerling et al., 1996*). We thus asked whether disease-relevant H3 mutants are sufficient to induce nuclear morphology defects. Histone H3.1 variants carrying dominant mutations K9M, or K27M or H3.3 mutants at K9M, K27M, or K36M were stably expressed in hTERT-immortalized fibroblast cells and nuclear morphology assessed. Expression of mutant histones in individual cells slightly increased expression of total histones but expression was not cytotoxic and did not affect proliferation as no changes to overall cell number were detected (*Figure 6—figure supplement 1*). Expression of histone H3.1 mutations had little effect on nuclear morphology (*Figure 6—figure supplement 2*). In contrast, expression of H3.3 mutants resulted in nuclear morphology changes and a

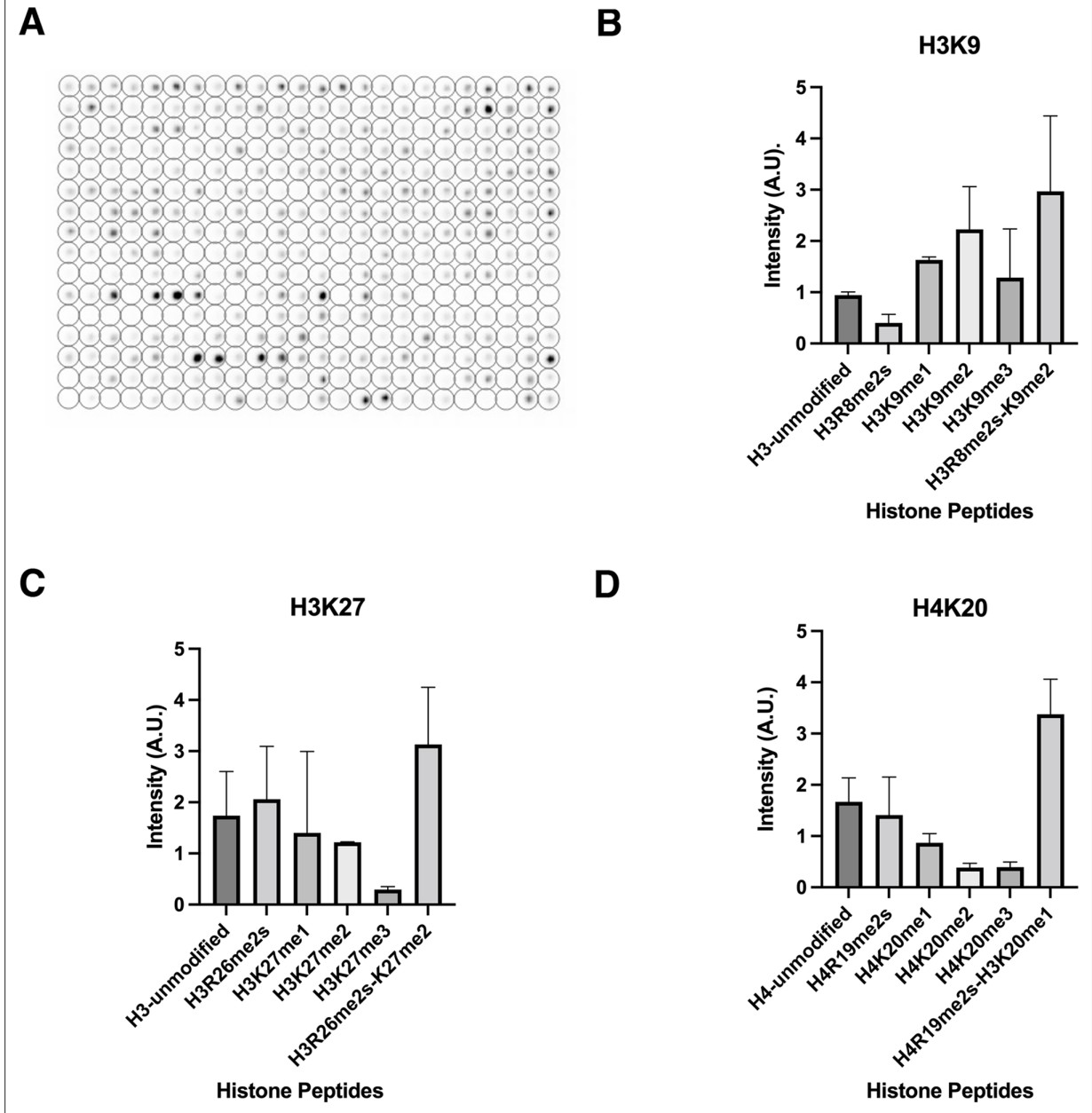

**Figure 5.** Specificity of lamin A binding histone modifications. (**A**) In vitro peptide binding array assay using GST-lamin A (506–646). Intensity of signal indicates binding. The peptide binding assay was performed three times. (**B**) Peptide binding assays for select histone H3K8/9 modifications. H3R8me2/K9me2 maintained the most intense signal compared to single modifications alone. Error bars indicate the standard deviation. (**C**) Peptide binding assay for histone H3R26/K27 modifications. H3R26me2/K27me2 maintained the most intense signal compared to single modifications alone. Error bars indicate the standard deviation. (**D**) Peptide binding assay for histone H4R19/K20 modifications. H4R19me2/K20me1 maintained the most intense signal compared to single modifications alone. Values represent intensity. Error bars indicate the standard deviation.

decrease in nuclear size (*Figure 6*). Expression of H3.3 mutants K9M, K27M, or K36M decreased circularity of the nucleus and reduced nuclear size (*Figure 6B–D*). Single cell analysis demonstrated that expression of the mutants increased the variability in size and shape in the population compared to expression of WT-H3.3, although no correlation between expression level of the mutant and morphological effects was observed (*Figure 6B–D*, *Figure 6—figure supplement 3*). In line with our observation of lack of correlation between lamin A levels and nuclear size and shape, while we find a slight increase in lamin A levels upon expression of either WT or mutant H3.3, the size and shape effects were specific to the mutants (*Figure 6—figure supplement 4*). No effect of H3.3 mutants on lamin

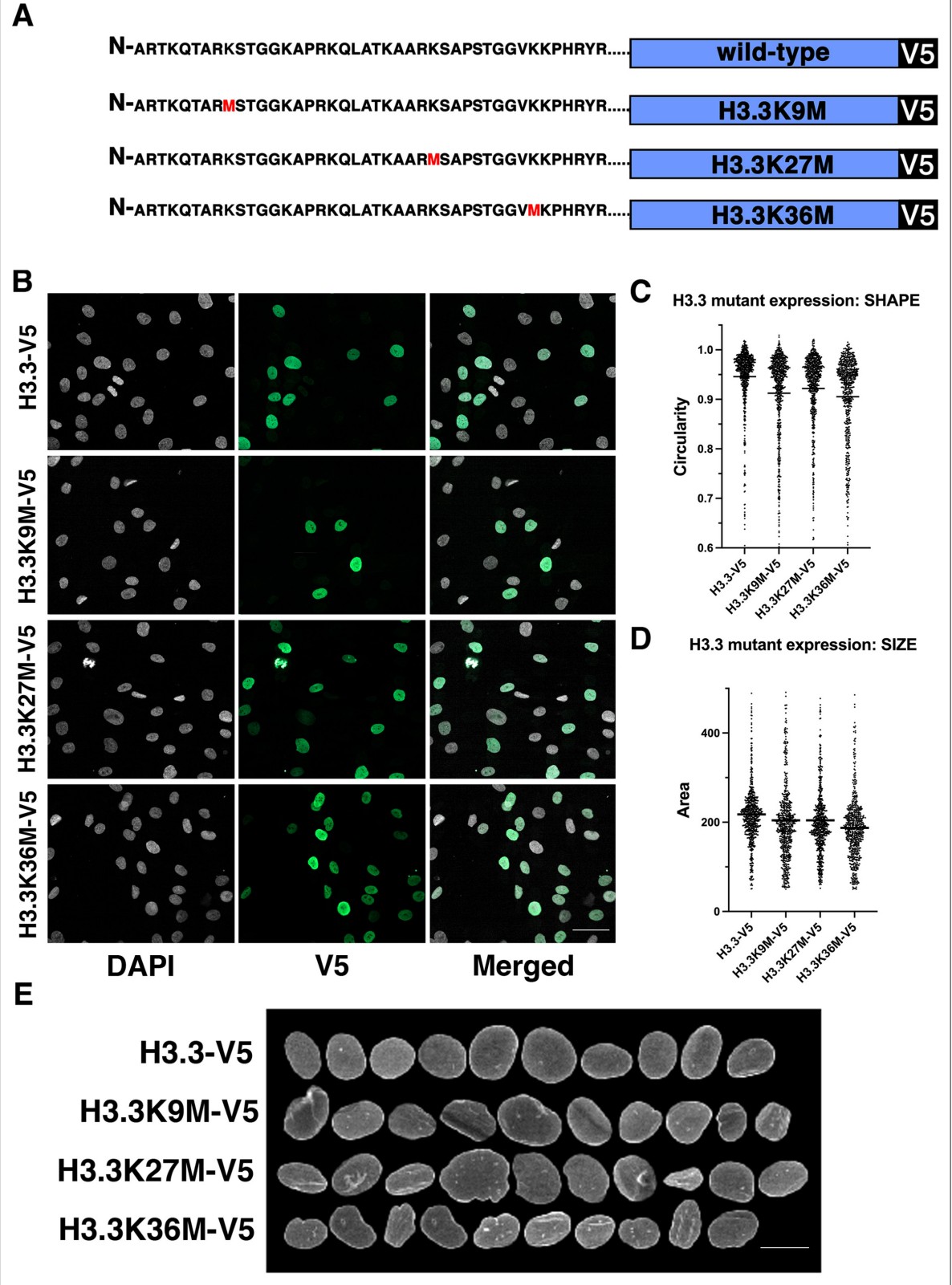

**Figure 6.** Expression of histone H3.3 mutants affect nuclear shape. (**A**) Wild-type (WT) and mutant histone expression constructs . Mutant histone expression experiments were performed with three biological replicates. (**B**) Stable expression of indicated H3.3 mutants in fibroblast cells. Gray: DAPI to detect DNA, green: V5-tagged histone variant. Scale bar = 50 µm. (**C**) Cells expressing histone H3.3 constructs reveal that histone H3.3K9, H3.3K27M, or H3.3K36M mutants showed reduced nuclear shape scores compared to WT H3.3 expression. The mean is indicated by the horizontal line. (**D**) Cells

*Figure 6 continued on next page*

*Figure 6 continued*

expressing histone H3.3 constructs reveal histone H3.3K9, H3.3K27M, or H3.3K36M mutants showed reduced nuclear size scores compared to WT H3.3 expression. The mean is indicated by the horizontal line. (**E**) Representative nuclei of cells expressing the indicated H3.3 variants. Signal represents lamin A staining. Scale bar = 10 μm.

The online version of this article includes the following figure supplement(s) for figure 6:

**Figure supplement 1.** Histone H3.3 total expression relative to wild-type H3.3-V5 and H3.3-V5 mutant expression.

**Figure supplement 2.** Histone H3.1 mutants display a lesser nuclear morphology phenotype compared to wild-type H3.1.

**Figure supplement 3.** Nuclear shape score relative to wild-type H3.3-V5 and H3.3-V5 mutant expression.

**Figure supplement 4.** Lamin A/C expression relative to wild-type H3.3-V5 and H3.3-V5 mutant expression.

**Figure supplement 5.** Single cell analysis of nuclear size and nuclear shape in cells expressing wild-type and mutant histone H3.3.

A localization was noted. We conclude that histone H3.3 mutations involved in lamin A interactions contribute to dysmorphia of the human cell nucleus.

## Discussion

Here, we have identified novel determinants of nuclear size and shape by utilizing an imaging-based functional genomics screen. Our findings highlight a prominent role of chromatin factors and epigenetic modifiers in the maintenance of nuclear morphology. In support of such a role, using in vitro binding assays, we find a direct interaction between lamin A and the modified tail of histone H3 and expression of disease-relevant histone H3.3 mutants altered normal nuclear morphology in human fibroblast cells.

Several cellular factors have previously been implicated in regulation of nuclear size and shape, including nucleocytoplasmic transport factors (*Jevtić et al., 2019*; *Levy and Heald, 2010*), components of the NPC (*Cantwell and Nurse, 2019*; *Iwamoto et al., 2009*; *Tamura and Hara-Nishimura, 2011*; *Ziubritskii and Slabinskii, 1991*), and nuclear envelope components (*Asencio et al., 2012*; *Bifulco et al., 2013*; *Cantwell and Nurse, 2019*; *Coffinier et al., 2011*; *Gant et al., 1999*; *Jevtić et al., 2015*; *Kume et al., 2019*; *Levy and Heald, 2012*; *Lu et al., 2012*; *Oda and Fukuda, 2011*; *Rowat et al., 2013*; *Wang et al., 2010*; *Zwerger et al., 2010*). In line with these earlier findings, we identified multiple components of the NPC such as *NUP205, NUP62, NUPL1,* and *NUP85* as well as a number of nuclear membrane proteins such as *SYNE1, CHMP6, TMEM19,* and *GOSR2* which validate our screening method. Our results are also in line with previous screening studies using the elliptic Fourier coefficient as a distinct parameter to quantitatively identify misshapen nuclei in MCF10A breast epithelial cells which targeted 608 epigenetic gene products and found 33 determinants of nuclear shape, including a number of epigenetic factors (*Tamashunas et al., 2020*). Interestingly, knockdown of some genes encoding core histones such as *HIST1H3B, HIST1H4B,* and *HIST1H2BA* also resulted in nuclear morphology defects (*Tamashunas et al., 2020*). Our analysis extends those studies by identifying several epigenetic factors, particularly histone modifiers and readers, as determinants of nuclear morphology. Furthermore, our experimental design assessed nuclear size in addition to nuclear shape in multiple cell types. Remarkably, we find distinct sets of size and shape determinants in individual cells lines. Lack of overlap between nuclear size and shape hits even among the same cell type underscores the complexity of nuclear morphology regulation and the need for large-scale screens using multiple measurement parameters in parallel to identify regulators of nuclear morphology.

Comparing the size and shape determinants in immortalized human fibroblasts and breast epithelial cells showed remarkably little overlap in determinants of nuclear morphology. The cell-type differences may be due to a number of reasons. One possibility is that different cell types use distinct networks and pathways to regulate nuclear morphology. Given that nuclear morphology does not seem to be controlled by a single dedicated pathway, but rather appears to be the result of multiple mechanisms, this scenario seems unlikely. Alternatively, it is possible that there are innate cellular features among cell lines that affect nuclear morphology. For example, differences in the rate of cell division, the amount of cell adhesion, nuclear import/export rates, or differing amounts of chromatin or lamin stability may affect nuclear morphology. Previously we found that knockdown of the nuclear pore component ELYS resulted in a aberrant nucleus phenotype in breast epithelial cells, and that comparison of nuclear size in ELYS knockdown cells among four different cell types found varying

degrees of nuclear size reduction (*Jevtić et al., 2019*). In the light of our finding of a prominent role of chromatin and epigenetic factors in determining nuclear morphology, an attractive possibility is that the differences in cellular factors that contribute to nuclear morphology in different cell types reflect cell type-specific epigenetic landscapes in which chromatin modulates nuclear morphology. More systematic analysis of a more diverse set of cell types in future studies should begin to address this question. While biologically distinct mechanisms likely regulate nuclear morphology and the nucleus-cell ratio, our screen focused on nuclear morphology given the technical challenges of accurately measuring the volumes of adherent cells. Other organisms such as fission yeast may be more amenable to screens for regulators of the nucleus-cell ratio (*Cantwell and Nurse, 2019*; *Kume et al., 2017*).

Lamins have been widely implicated in maintenance of nuclear morphology (*Lammerding et al., 2004*; *Matias et al., 2022*). Reassuringly, and as expected, *LMNA* was the second strongest hit in our screens. However, many effectors of size and shape identified here exerted their effect without affecting lamin A/C levels. Instead, the nuclear shape screen identified a group of chromatin modifiers. This is in line with previous work pointing to a combined contribution of lamins and chromatin, and their interplay, to nuclear morphology and biophysical properties (*Stephens et al., 2017*). A prominent role of chromatin in nuclear morphology is suggested by the observation that nuclear blebbing can be promoted or inhibited by treating cells with drugs that increase euchromatin or heterochromatin, respectively (*Stephens et al., 2018*). In addition, alterations to euchromatin and heterochromatin rescue nuclear morphology defects in disease model cells (*Stephens et al., 2018*). Furthermore, the lysine acetyltransferase HAT1 which acts on newly incorporated histone H4 increased nuclear size, nuclear blebbing, and micronuclei and loss of HAT1 acetylation disrupts chromatin regions associated with the nuclear lamina (*Popova et al., 2021*). Along the same lines, acetylation of lamin A via the acetyltransferase MOF leads to changes in nuclear morphology and epigenetic alterations (*Karoutas et al., 2019*). These observations, combined with our findings, highlight a prominent role for histone modifications in regulation of nuclear size and shape. Our finding of a direct effect of histone modification mutants of H3.3 on nuclear morphology supports this scenario.

Although it is well established that chromatin is closely juxtaposed with the nuclear lamina and genome regions which associate with the lamina can be mapped as lamin-associated domains, the precise nature of chromatin-lamin interactions is largely unknown. We find that lamin A, but not lamin C, directly interacts with histone H3. This finding adds to the prior identification of a C-terminal region, present on both lamin A and lamin C, that can bind to DNA, and linker DNA assembled onto nucleosomes (*Stierlé et al., 2003*). Furthermore, progerin mutations reduced lamin-DNA interactions (*Bruston et al., 2010*). Our studies identify a novel lamin A-histone H3 interaction independent of DNA binding. We map two distinct regions located within the C-terminal unstructured tail and within the globular domain which are required for lamin A-H3 interactions, and we suggest that these interactions occur in the context of lamin A monomers. The binding of lamin A to histones seems to be facilitated by histone modifications, because we find enhanced binding of lamin A to dual (Rme2/Kme1-me2) modifications which are associated with transcriptionally repressive marks on heterochromatin (*Di Lorenzo and Bedford, 2011*; *Zhang and Reinberg, 2001*). These findings point to a mechanism by which chromatin-lamin A interactions via modified histone H3 tails contribute significantly to nuclear morphology. While our results provide information on the in vitro binding of lamin A and histones, the role and regulatory mechanisms of lamin A in binding to either DNA or histones in vivo, and its subsequent recruitment to chromatin in the complex in vivo nuclear environment, remain to be explored. The precise nature of this interaction and what the downstream effect of this lamin A-histone H3 interaction is will require further studies.

Changes to nuclear shape and size have been documented in many types of cancer and nuclear morphology changes are often correlative with poor prognosis (*Pienta and Coffey, 1991*; *Wolberg et al., 1999*; *Zink et al., 2004*). In line with a role of disease-associated histone epigenetic modifiers in contributing to nuclear dysmorphia and disease, we find that expression of K9 and K27 methylation mutants of histone H3.3, in which K27 mutants are also oncogenic, shows changes to nuclear morphology. This finding is relevant since histone H3.3K27M mutations were identified in a subset of pediatric patients with glioblastoma (*Khuong-Quang et al., 2012*; *Schwartzentruber et al., 2012*) and H3.3K36M mutations were documented in chondrosarcomas (*Behjati et al., 2013*), which are also

characterized by extensive nuclear aberrations. Although we do not know whether oncohistones exert their effects via lamin A and/or changes in nuclear size and shape, these findings are in line with our observation of direct physical interaction of lamins with histones.

Taken together, the use of imaging-based screening reported here significantly expands the list of cellular factors that contribute to nuclear morphology. Our findings of an enrichment of chromatin factors and the fact that the vast majority of nuclear size and shape effectors exerts their function without alteration of lamin protein levels highlight the important role chromatin plays in determining nuclear shape and size.

## Materials and methods

### Human cell culture

Previously described karyotypically normal hTERT-immortalized dermal fibroblasts cells (*Scaffidi and Misteli, 2011*) were maintained at 37°C with 5% $CO_2$ in minimum essential medium containing 15% fetal bovine serum, 100 U/mL penicillin, 100 µg/mL streptomycin, 2 mM L-glutamine, and 1 mM sodium pyruvate. MCF10AT1k.cl2 cells (Barbara Ann Karmanos Cancer Institute) (*Dawson et al., 1996*; *Heppner and Wolman, 1999*) were cultured at 37°C with 5% $CO_2$ in DMEM/F12 media supplemented with 1 mM $CaCl_2$, 5% horse serum, 10 mM HEPES, 10 µg/mL insulin, 20 ng/mL EGF, 0.5 µg/mL hydrocortisone, and 0.1 µg/mL cholera toxin. Cell lines are available upon request. Validated cell lines CRL-1474 (ATTC; RRID: ACVCL_2384; https://www.cellosaurus.org/CVCL_2384) and MCF10AT1k.cl2 (Karmanos Cancer Institute Repository; RRID: CVCL_WM98; https://www.cellosaurus.org/CVCL_WM98) were used. Cell lines were periodically tested for mycoplasma.

### High-throughput screen

For high-throughput siRNA screening, 1200 cells were seeded into each well of a CellCarrier-384 Ultra microplate (PerkinElmer) using a Multidrop Combi Reagent Dispenser (Thermo Fisher). Cells were reverse transfected with siRNA oligos targeting specific genes at a 20 nM final concentration in 40 µL of complete media and were grown for 72 hr at 37°C. The screen used custom siRNA libraries (siRNA Silencer Select, Thermo Fisher) targeting proteins that localize to the nuclear membrane (346 genes) or proteins involved in epigenetic and chromatin regulation (521 genes) (*Supplementary file 1A*). Each gene was targeted with three different siRNAs placed in individual wells, for a total of 2601 siRNA experiments per screen. A non-targeting, scrambled siRNA (Thermo Fisher, #4390847) was used as a negative control and the AllStars Hs Cell Death Control siRNA (QIAGEN) was used as a control to score transfection efficiency and for assay optimization. An siRNA targeting lamin A/C (Thermo Fisher, #8390824) was used as a positive biological control to score nuclear shape changes. After siRNA treatment, cells were fixed in 4% paraformaldehyde (PFA) in PBS for 20 min at room temperature, washed three times for 5 min in PBS, permeabilized with 0.5% Triton X-100 in PBS for 15 min, washed three times for 5 min in PBS, and blocked in PBS with 0.05% Tween 20 (PBST) and 5% BSA for 30 min. For detection and measurement of nuclear morphology, cells were immuno-stained with primary antibodies against lamin A/C (Santa Cruz, sc-376248, mouse, 1:1000) and lamin B1 (Santa Cruz, sc-6217, goat, 1:500) in PBST with 1% BSA for 4 hr at room temperature or overnight at 4°C. Cells were then washed three times for 5 min with PBST and incubated for 1 hr at room temperature with secondary antibodies diluted in 1% BSA in PBST containing DAPI (5 ng/µL), before washing three times for 5 min in PBST. The screen was performed in two biological replicates on different days.

Immuno-stained plates were imaged using an Opera QEHS (PerkinElmer) dual spinning disk high-throughput confocal microscope using a 40× water immersion lens (NA 0.9). The high-throughput microscope acquired images using two CCD cameras (1.3 Megapixels) with pixel binning set at 2×2 (pixel size: 323 nm). The DAPI channel utilized the 405 nm laser for excitation and the 450/50 nm bandpass filter for acquisition, lamin B1 expression was imaged using the 488 nm laser for excitation and a 520/35 nm bandpass filter for acquisition, and lamin A/C expression was imaged using the 561 nm laser for excitation and the 600/40 nm bandpass filter for acquisition. DAPI, lamin B1, and lamin A/C images were acquired at a single focal plane. Thirty randomly selected fields of view were imaged per well, and typically >250 cells per well were analyzed.

Quantification of nuclear size and shape were performed using a customized software pipeline. Images generated were analyzed using the Columbus 2.6 high content imaging analysis software

(PerkinElmer). An analysis pipeline was generated where nuclei were segmented using the DAPI staining image. Partial nuclei located at the edge of the image were excluded from subsequent steps of the analysis. To identify nuclear geometric changes, nuclear area, width, length, and circumference were measured as well as mean fluorescence intensity of lamin A/C and lamin B1. Single cell measurements were computed into mean per well values. To identify nuclear shape hits, circularity values (circularity = $4\pi$Area/perimeter$^2$) were calculated on a mean per well basis. To identify nuclear size hits, nuclear area was calculated on a mean per well basis. RStudio software, R, and the cellHTS2 R package (v 2.36.0) (*Boutros et al., 2006*) used the B-score method (*Brideau et al., 2003*) to normalize mean per well values on a per plate basis using the median of all the library wells on the plate. B-score values for all the samples in a single biological replicate were then normalized across a single replicated to generate Z-scores (*Figure 1—figure supplement 2*) for each well/siRNA oligo.

The Z-score value for each sample (well) was calculated as the robust Z-score = (B-score value – median of the B-score for all samples)/(median absolute deviation of the B-scores for the samples). The Z-score indicates how many units of variance, either above or below, a sample is from the median of all measurements for the samples. For an individual sample, the further the Z-score is from 0, the larger the degree in which the sample is from the median of all samples. A diagram explaining the Z-score calculations used to measure nuclear size and shape is presented in *Figure 1—figure supplement 2*. The screen was performed twice on different days, and Z-scores from each biological replicate were averaged to generate a final mean Z-score for each siRNA oligo in the library. Positive hit genes were identified by Z-scores ±1.5 in at least two of the three unique siRNAs targeting each gene. Hits that fit these criteria but that displayed a significant effect (Z-score equal or less than –2) on cell number were eliminated from analysis and were confirmed to include several known regulators of the cell cycle, proliferation, and mitotic progression (*Figure 1—figure supplement 3*).

## Recombinant protein expression and purification

Fragments of human lamin A, lamin C, and progerin were cloned into the pGEX4T1 vector to generate N-terminally tagged GST fusion proteins. The plasmids used for recombinant protein production are listed in *Supplementary file 1B*. Proteins were expressed in *Escherichia coli* Rosetta 2 (Novagen) in LB media. Protein expression was induced by the addition of 0.2 mM IPTG for 18 hr at 18°C. Cells were collected and suspended in lysis buffer containing 50 mM Tris pH 7.5, 150 mM NaCl, 0.05% NP-40, 1 mM PMSF, protease inhibitors, and 0.5 mg/mL lysozyme. Cells were incubated for 30 min on ice and lysed and sonicated for a duration of 20 s at 18% amplitude using a Branson Digital Sonifier 250 with a 102C converter set. Lysed cells were centrifuged at 21, 000 × *g* at 4°C for 15 min. The supernatant was removed and incubated with glutathione Sepharose 4B resin (GE). Beads were washed twice with lysis buffer and once with elution buffer (100 mM Tris-HCl, pH 8.0). Recombinant GST fusion proteins were eluted by resuspending the resin in elution buffer containing 15 mg/mL reduced L-glutathione (Sigma) and incubated at 4°C for 4 hr. Recombinant proteins were run on a 4–12% BisTris gel and analyzed using colloidal staining. Plasmids are available upon request.

## Calf thymus histone binding assay

Calf thymus histone binding assays were performed by incubating 50 µg of calf thymus histones (Worthington) with 10 µg of purified GST fusion proteins in binding buffer containing 50 mM Tris pH 7.5, 1 M NaCl, and 1% NP-40 overnight at 4°C. To identify histone-lamin interactions, glutathione Sepharose 4B resin (GE) was added for 1 hr. Beads were washed five times in binding buffer, resuspended in 4× Laemmli buffer, run on a 4–12% BisTris gel, and transferred to a PDVF membrane. Membranes were used for western blot analysis using the antibodies at the specified concentrations (*Supplementary file 1C*). Purifications of recombinant proteins and calf thymus histone binding experiments were performed at least three times.

## Peptide array binding assay

MODified histone peptide arrays (Active Motif) composed of 384 unique histone peptides representing acetylation, citrullination, methylation, and phosphorylation post-translational modifications were blocked with TBST (10 mM Tris/HCl pH 7.5, 0.05% Tween-20, and 150 mM NaCl) and 5% nonfat milk overnight at 4°C. The peptide array was washed twice with TBST, one time with interaction buffer (100 mM KCl, 20 mM HEPES pH 7.5, 1 mM EDTA, 0.1 mM DTT, and 10% glycerol). The array

was incubated with 10 nM purified GST-lamin A (pACS37) in interaction buffer at room temperature for 1 hr. The array was then washed three times in TBST, and incubated with anti-GST (GE, #27-4577-01, 1:5000) for 1 hr at room temperature in TBST with 1% nonfat dried milk. The array was further washed three times in TBST with 10 min for each wash and incubated with HRP-conjugated secondary antibody (Santa Cruz) for 1 hr at room temperature. The membrane was submerged in ECL solution (Amersham), imaged, and intensity data was quantified using Array Analyzer Software (Active Motif). Purifications of recombinant proteins and peptide binding experiments were performed at least three times.

### Expression of histone mutants in human cells

Lentiviruses containing wild-type histone H3.1 and H3.3 histone constructs were generated with the pLenti6.3/V5-TOPO TA Cloning kit (Thermo Fisher). Mutations were added to wild-type H3.1-V5 and H3.3-V5 sequences using the Quikchange II XL site directed mutagenesis kit (Agilent). Lentiviruses were transfected into HEK293T cells in combination with viral packing and envelope plasmids pSPAX (Addgene 12260) and pMD2.G (Addgene 12259), respectively, and allowed to incubate for 48 hr before virus harvesting. Viral supernatant was collected from the HEK293T cells and placed onto hTERT-immortalized fibroblast cells and incubated for 24 hr. Fibroblast cells were selected for mutant histone expression by the addition of 5 μg/mL of blasticidin for 10 days, 1200 cells were seeded onto 384-well plates, and were grown for 72 hr at 37°C. After treatment, cells were fixed in 4% PFA in PBS for 20 min at room temperature, washed three times for 5 min in PBS, permeabilized with 0.5% Triton X-100 in PBS or 15 min, washed three times for 5 min in PBS, and blocked in PBS with 0.05% Tween 20 (PBST) and 5% BSA for 30 min. For detection and measurement of nuclear morphology, cells were immuno-stained with primary antibodies against lamin A/C (Santa Cruz, sc-376248, mouse, 1:1000) and lamin B1 (Santa Cruz, sc-6217, goat, 1:500) (*Supplementary file 1C*) in PBST with 1% BSA for 4 hr at room temperature or overnight at 4°C. Cells were washed three times for 5 min with PBST and incubated for 1 hr at room temperature with secondary antibodies diluted in 1% BSA in PBST containing DAPI (5 ng/μL). Cells were washed three times for 5 min in PBST. Cells were imaged using a CV7000 high-throughput spinning disk confocal microscope (Yokogawa) with a 20× air objective (NA 0.75) and two sCMOS 2550×2160 pixel (5.5 Megapixel) cameras. Images were binned 2X2 (pixel size: 650 nm). Images taken on the CV7000 microscope were analyzed using Columbus 2.8.1 software (PerkinElmer). A Columbus image analysis pipeline segmented nuclei using the DAPI image, and then measured nuclear parameters: nuclear area, width, length, circumference, and the mean intensity levels of stained proteins.

### Statistical analysis

Scatterplots displaying the relationship between two measurements were assessed using Spearman's coefficient analysis. Nuclear morphology features in cells expressing mutant histones were analyzed by using a two-sample Kolmogorov-Smirnov test to compare the distribution of nuclear morphology scores at the single cell level.

## Acknowledgements

We thank Drs. Leonard Kubben (IMB, Mainz), Sigal Shachar (Arcellx), and Akanksha Singh (Active Motif) for advice, protocols, and troubleshooting. Work in the Misteli lab and at HiTIF was supported by the Intramural Research Program of the NIH, NCI, Center for Cancer Research via 1-ZIA-BC010309-23 and 1-ZIC-BC011567-08, respectively. Work in the Levy lab was supported by the National Institutes of Health/National Institute of General Medical Sciences (R35GM134885 and P20GM103432) and the USDA National Institute of Food and Agriculture (Hatch project #1012152).

## Additional information

### Funding

| Funder | Grant reference number | Author |
|---|---|---|
| National Institutes of Health | NIH 1-ZIA-BC010309-23 | Tom Misteli |
| National Institutes of Health | NIH 1-ZIC-BC011567-08 | Gianluca Pegoraro |
| National Institutes of Health | NIH R35GM134885 | Daniel L Levy |
| National Institutes of Health | NIH P20GM103432 | Daniel L Levy |
| USDA National Institute of Food and Agriculture | Hatch project #1012152 | Daniel L Levy |

The funders had no role in study design, data collection and interpretation, or the decision to submit the work for publication.

### Author contributions

Andria C Schibler, Data curation, Writing – original draft; Predrag Jevtic, Data curation; Gianluca Pegoraro, Resources, Software, Visualization, Methodology; Daniel L Levy, Tom Misteli, Conceptualization

### Author ORCIDs

Predrag Jevtic 
Gianluca Pegoraro 
Daniel L Levy 
Tom Misteli 

### Decision letter and Author response

Decision letter https://doi.org/10.7554/eLife.80653.sa1
Author response https://doi.org/10.7554/eLife.80653.sa2

## Additional files

### Supplementary files

• Supplementary file 1. High-throughput screening targets and hits. (A) lists the genes targeted in the screen. (B) is a file describing the plasmids generated during this study. (C) lists the antibodies used in this study. (D) is a comparative list of nuclear shape Z-score, nuclear shape raw score, nuclear area, and nuclear perimeter measurements.

• Supplementary file 2. Nuclear shape hits. (A) lists hits altering nuclear shape in fibroblast cells. (B) is a list of hits resulting in lower lamin A/C expression. (C) is a list of hits resulting in lowered lamin B1 expression.

• Supplementary file 3. Screen validation. (A) lists validation results for the nuclear shape screen in fibroblast cells. (B) lists validation results for hits increasing nuclear size in fibroblast cells. (C) lists validation results for hits decreasing nuclear size in fibroblast cells. (D) identifies nuclear shape hits in MCF10AT cells. (E) lists lamin A interacting peptides.

• MDAR checklist

### Data availability

All data and materials generated in this study were placed in a repository or available upon request. Datasets generated from the high-throughput screen and validation assays have been deposited at GitHub – https://github.com/CBIIT/mistelilab-nucleus-size-shape-screen, copy archived at *Pegoraro, 2023*. Source data files used in figures have been provided.

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
