## [Editor Report]

In this manuscript the authors describe targeted, imaging-based RNAi screens to identify novel modulators of nuclear size and shape, which are established diagnostic and prognostic indicators of human diseases including cancer. This work provides new insights into the molecules that dictate nuclear morphology tied to chromatin state, the nuclear lamina, and the nuclear envelope. This resource will be broadly valuable to the nuclear cell biology and chromatin biology fields.

---

## [Decision Letter]

**Decision letter after peer review:**

Thank you for submitting your article "Identification of epigenetic modulators as determinants of nuclear size and shape" for consideration by *eLife*. Your article has been reviewed by 3 peer reviewers, one of whom is a member of our Board of Reviewing Editors, and the evaluation has been overseen by and Jessica Tyler as the Senior Editor. The following individual involved in review of your submission has agreed to reveal their identity: Dennis Discher (Reviewer #3).

Essential Revisions:

1. Transparency and interpretability of how the screen was performed and the use of Z-scores: The authors must provide a clear explanation of the z-score and how it is produced from the raw data to represent nuclear roundness/shape and size in the manuscript. In particular, there was concern that the measure might not address whether there are relatively few nuclei with very perturbed structures or instead a rather smaller deviation across all or most cells in a given condition – more clarity on this should be included in the revision. The authors are also encouraged to provide values or examples along the z-score axis to allow the reader to see how the direct measurements (area and perimeter) translate into the reported z-scores. Addressing these points is essential for readers to put this study in the context of others that report data as a deviation from circularity from 1 to 0, as well as allowing the reader to compare data in Figures1-3 and Figure 6. Last, the authors need to more clearly justify and communicate that they have removed hits with z-scores below -2, including stating that the focus of the work is on what amounts to genes with more subtle effects on nuclear size and shape. Given the stated criteria, it is also important for the authors to address how they have concluded that the included screen hits do not impact the cell cycle and cell viability.

2 Statistics and robustness: The authors need to provide more detail on how they have justified that two replicates are sufficient for the reported data. For example, are they leveraging that they have employed 3 unique siRNAs? It would also build confidence if a subset of hits discussed in detail were tested for the knock-down efficiency and its relationship to nuclear appearance.

3. Interpretation of the screen "hits": As the screen was already biased towards nuclear envelope and chromatin factors, an unbiased approach to further interpret the hits would strengthen the overall message – for example, an approach that can identify factors that interact with one another, indicating a role for a complex, etc. – many such tools are readily available. At a minimum, the authors should provide greater context, particularly for the hits that they decide to elaborate on.

4. The need to temper claims about distinct sensitivities of cell types to perturbations that alter nuclear appearance or to provide additional experimental support: The conclusions made from the comparison of the hits from the two independent screens without further testing were found to be inadequately supported. The authors must test a subset of the hits from one cell line in the other (and vice versa) in order to demonstrate that there are indeed cell type specific perturbations. The authors should also compare their list to other published works.

5. Interpretation of the biochemistry: There were several questions about the interpretation of the GST-lamin A biochemistry that need to be addressed. First, it is well established that GST imposes dimerization on proteins expressed as GST fusions independent of cysteines – the findings need to be reconsidered in this light, particularly as it relates to the combined lamin A/lamin C experiments. Second, even in the case that disulfide bonds between cysteines are occurring, there needs to be further consideration of whether this is possibly relevant in vivo (in the reducing environment of the nucleus) and/or requires further characterization, specifically evidence that it is indeed disulfide bonds between the lamin A and not the GST. Third, additional quantitative context for the binding of lamin A to the modified histone tails is needed (are the interactions saturable, what is the realm of the Kd) – as only relative values are provided in Figure 5, the reader currently lacks a framework for knowing what the order of magnitude is for these interactions.

6. Need for more depth in the analysis of histone H3.3 variant effects (Figure 6): As these mutants are expressed over WT histones the authors should address how expression level impacts the measured effects on nuclear appearance. What might the ratio of H3.3-V5 to H3.3 be in these experiments (this should be addressable due to the increases size of the V5 on a small histone). In single cells based on the immunostaining, the authors can also address whether there are trends according to the expression level. In this analysis, please also clarify if "shape" is independent of a change in area. There is also a need for statistical analysis. Last, please provide further evidence that the H3 effects manifest through lamins – (also relates to former figures) through localization, relationships to genomic data, etc.

*Reviewer #1 (Recommendations for the authors):*

1. In the introduction of evidence tying nuclear size to nuclear transport additional studies should be referenced in particular the effects of disrupting nuclear export with the small molecule leptomycin B (validated, in fact, by its target XPO1 in the size screen).

2. Please clarify if the lamin A fragment includes the C-terminus after processing or not.

3. The introduction to experiments included in Figure 5 needs to state what form of lamin is being interrogated.

4. The authors must cite explicit examples with references when suggesting that the presence of hits including nucleoporins and nuclear membrane proteins "validates their screening method".

*Reviewer #2 (Recommendations for the authors):*

This work is a fantastic advance in the field that provides many novel and deeply interesting findings. However, the paper's large and meaningful amount of data that is completely confused using Z scores (not raw data) and how to deal with variation (no error bars). These two major criticisms of the paper should be addressed before the paper is published.

1. There needs to be a clear explanation of the z-score and how it is produced from data for nuclear roundness/shape and size. Overall, the paper would benefit greatly from graphing roundness and size data next to z score to allow the reader to see how the direct measurements (area and perimeter) translate into z score, the measure of change from the mean.

a. The major problem with this paper lies in how the bulk data is scored and displayed. Z score is not explained anywhere causing major confusion for the reader.

b. The core measurement of this paper is area 4 π divided by perimeter squared which is not shown in the paper in any capacity relative to the Z store. It would be gratefully helpful to provide at least one if not a few examples of the type of average or distribution of roundness scores relative to the output Z score. This is important as many previous papers in this field report shape measurements at 1 being a perfect circle and abnormal nuclear morphology measure along this scale from 1 to 0. While abnormal shapes are shown – there connection to the reported Z score and roundness value underlying it is hidden. Same is true for size where exact numbers would be useful to the reader.

i. Please show some amount of raw comparative roundness and size measurements.

ii. Why not show the > 250 nuclear measurements for scrambled, LMNA, and a few choice hits across the scale of z score?

c. While the first half of the paper reports Z scores (Figure 1-3), later in the paper roundness scores are reported for Figure 6. This is confusing for the reader as measurement of change in nuclear shape and size changes in the paper. This example highlights the importance of being able to bridge the use of these two values, why they are used at different times, and overall understanding. The later histogram like distributions of roundness have no clear statistical measure from the figure.

d. THE SPECIFIC question centers around how much the roundness value differs between hits and how much roundness and size vary within one hit. In diagnostic tests of human diseases abnormal nuclear morphology percentages can change by very little between non-aggressive and very aggressive illnesses.

i. How much does Z score capture this vs ignore a sub population of abnormally shaped nuclei?

e. It is appreciated that raw data were made available through GitHub. However, the problem with many raw data sets is that it is hard to find the numbers that matter in a meaningful way.

2. The manuscript should clearly state why two replicates underscore the majority of the data and that the data graphs provide no error bars.

a. The manuscript should justify why two replicates are sufficient for most of the reported data. If you can do two replicates why not have done three to provide the agreed upon triplicate measurement.

i. The fact that 3 unique siRNAs are used for each KD is not discussed. Does this provide multiple replicates for the paper that are not discussed?

b. For z scores, the paper clearly reports two replicates but how are those two replicates are used in the data. The variation from one experiment to the other is not clearly provided. Furthermore, with no triplicate measure should a small population of hits be verified in a more detailed manner like is common in many gene screens? Again, showing just the roundness or size measure would be highly informative.

c. For binding values, there are also no error bars. How is the reader supposed to interpret the data with no idea of variance?

d. Figure 6 C and D is written in a cryptic manner. Does the manuscript mean to say that compared to wild-type H3.3 all mutant forms (list them) show statistically significant changes in shape C and size D? or are there differences between different mutants? It is unclear.

*Reviewer #3 (Recommendations for the authors):*

1. The screening of Figures1-3 is fine, but I found it important that they restricted analyses "To eliminate hits due to cell death or altered cell-cycle behavior, we excluded any hits with a cell number z-score of less than -2." Some mention of this in the abstract seems important. For example: 'The most dramatic effects were found for cell cycle and death regulators, but we chose to focus on regulators of more subtle effects.' In this regard, the authors need to take great care that the targets identified have zero impact on cell cycle and death.

2. The authors add "reducing agent DTT to inhibit dimerization" of GST-lamin A, but they need to tell us the specific Cys-Cys crossbridges that cause dimerization because GST also has Cys and could be contributing to non-physiological oligomers.

3. In analyzing laminA-histone interactions, the authors write "For example, GST-laminA bound ~ 5-fold more efficiently to histone H3R8me2s/K9me2 than histone H3K9me2." The word "efficiently" does not seem a standard descriptor for binding, compared to terms like affinity, specificity, saturation, etc. The authors should strive to show at least one of the 'stronger' interactions is saturable and has a reasonable Kd as a basis for specificity.

4. The histone-H3 mutation effects on nuclear morphology in Figure 6 are important, but some key details and insights are needed. These seem to be Lenti's that transduced a fraction of the fibroblast cultures, giving levels that are zero, low, or very high in each culture, but does that variation have any effect? Histone levels do relate to cell cycle (e.g. PMID: 35760914), and so are high expressers at later stages of cell cycle (e.g. higher DNA staining) and would that be sensible for how the construct is regulated in its expression? Does the 'shape' parameter neglect differences in 'area'. Are the histone intensities uniform, and what happens to LaminA levels or localization?

---

## [Author Response]

Essential Revisions:1. Transparency and interpretability of how the screen was performed and the use of Z-scores: The authors must provide a clear explanation of the z-score and how it is produced from the raw data to represent nuclear roundness/shape and size in the manuscript.

As requested, we now explain in a new Figure 1 —figure supplement 2 in detail how Z-scores were derived from the raw data. In addition, we further describe and clarify statistical methods in the Materials and methods and in Results sections on p. 7, 8, 9, and 22.

We now explicitly explain that the Z-score is a standard statistical measure used in screening datasets and describes how different an individual sample in the screen is relative to the mean of all samples in the screen PMID: 16869968. For a publicly available standard definition see here. Additional details are provided in the documentation of the R package , used to calculate all the statistics for the screens in this manuscript. We calculated separate Z-scores for either shape or size for each well in the screen using the geometrical measurements as described in the Materials and methods and Results.

More specifically, to obtain a roundness/shape parameter, the circularity of each imaged nucleus was measured using the 4pArea/perimeter^2^ formula. To obtain a size parameter for the nucleus the area of each imaged nucleus was first measured. Raw per nucleus values for all measured parameters were then averaged on a per well (oligo siRNA) basis. Mean per well values were then normalized on a per plate basis. Z-scores for two biological replicates were averaged, thus providing one Z-score values per oligo siRNA. Finally, since each gene is targeted by multiple siRNA oligos in the library, we picked the median of the Z-scores for siRNA oligos against a gene (equivalent to the oligo with 2^nd^ out of 3 strongest biological effect in the assay). This is the Z-score value reported in the manuscript to score hits in the primary screen, and it measures the relative strength of the nuclear morphology change between the knock-out of a particular gene and the median of the population (e.g. Z-score = 0). This approach is standard in the analysis of RNAi screens (see PMID: 16869968). The analysis workflow detailed above is now presented in Figure 1 —figure supplement 2, Figure 1A and is described in detail on p. 7, 8, 9, and 22.

In particular, there was concern that the measure might not address whether there are relatively few nuclei with very perturbed structures or instead a rather smaller deviation across all or most cells in a given condition – more clarity on this should be included in the revision.

As most of our studies rely on single cell analysis, we very much appreciate this point. To further analyze the data from the initial screen, we now provide, as requested, a new Figure 1 —figure supplement 5. Figure 1 —figure supplement 5 which contains single cell data for nuclear shape values of several hits and controls in histogram format and compare distributions to control cells. We find that the Z-score values on a per well/siRNA/gene basis are not driven by alterations in small subpopulations of cells, but rather reflect changes to nuclear morphology present in most cells in the population as indicated by the shift of the entire distribution. We now mention these data on p. 8.

The authors are also encouraged to provide values or examples along the z-score axis to allow the reader to see how the direct measurements (area and perimeter) translate into the reported z-scores. Addressing these points is essential for readers to put this study in the context of others that report data as a deviation from circularity from 1 to 0, as well as allowing the reader to compare data in Figures1-3 and Figure 6.

As requested, Z-score values and raw circularity values have now been added to figures 1. Furthermore, Figure 1 —figure supplement 4 was added to give a sense of the relationship of phenotypic measurements of area, perimeter, and circularity in comparison to Z-scores generated from the screen. In addition, a new Supplementary File 1D with the Z-score next to the raw mean per well circularity score is now included for comparison and mentioned on page 7. The reason for the use of well/siRNA/gene Z-scores in Figure 1 and 2 is because these figures represent screening data, whereas Figure 6 represents single knockdown experiments to which Z-scores cannot be applied since Z-scores compare a sample to all other samples within a screening dataset.

Last, the authors need to more clearly justify and communicate that they have removed hits with z-scores below -2, including stating that the focus of the work is on what amounts to genes with more subtle effects on nuclear size and shape. Given the stated criteria, it is also important for the authors to address how they have concluded that the included screen hits do not impact the cell cycle and cell viability.

We agree this is very important and we now mention this caveat in the abstract as suggested by the referee. We found that some hits in our initial shape screen were known regulators of the cell cycle and of mitosis such as *AURKB, MAD2L1* and *CDC2* and displayed a low cell count per well compared to control cells. To eliminate potential false positives, we considered the resulting nuclear morphology changes as secondary effects due to cellular stress, abnormal cell division, siRNA toxicity, and potential lethality as they were often accompanied by significant reduction of cell number. Therefore, we removed nuclear shape and size hits from further analysis if their cell number per well Z-score was below -2. Nuclear shape hits with Z-score below -1.5, described in Figure 1, and no significant changes in cell number, were included in our hit list. We now describe in more detail these criteria and our strategy to reduce false positives and artifacts in the Materials and methods on p. 22 and in the Results sections on p.8.

2 Statistics and robustness: The authors need to provide more detail on how they have justified that two replicates are sufficient for the reported data. For example, are they leveraging that they have employed 3 unique siRNAs? It would also build confidence if a subset of hits discussed in detail were tested for the knock-down efficiency and its relationship to nuclear appearance.

As requested, we have added a more detailed description of statistical methods and robustness on p. 21 and 22. As the referee points out, we use three separate siRNAs for each target gene and hits are defined by an effect of at least 2 of the 3 siRNAs, reducing the likelihood of off target effects. It is a standard practice in the field to perform large scale screens such as the ones reported here in duplicate, especially when multiple siRNAs are used in an unpooled fashion as done here (PMID: 19644458). A comparison of the two replicates of the screen indicated strong correlation very good correspondence between hit lists (see Figure 1 —figure supplement 1) and positive hits from the primary screen were validated in secondary screens which are now shown in Figure 3 —figure supplement 6. Furthermore, the generated datasets are data-rich and based on several hundred images and involve analysis of typically 500-1000 cells per sample, generating highly robust datasets. To further address this point, we have now included, as requested, Supplementary Files 3A, 3B, and 3C, which detail the results of a validation screen using siRNAs with differing target sequences and generated using different chemistry. We find good correspondence between primary hits and our validation screen. Additionally, as requested, we now build confidence in our hit identification by inclusion of new data in Figure 3 —figure supplement 2, where we repeated knockdown experiments and used antibody staining and automated imaging-based quantification of fluorescence intensity to demonstrate high knock-down efficiency of several hits. We find knockdown efficiency to be extremely robust with typical reduction levels of 60-90% using 8 different antibodies (Figure 3 —figure supplement 2).

3. Interpretation of the screen "hits": As the screen was already biased towards nuclear envelope and chromatin factors, an unbiased approach to further interpret the hits would strengthen the overall message – for example, an approach that can identify factors that interact with one another, indicating a role for a complex, etc. – many such tools are readily available. At a minimum, the authors should provide greater context, particularly for the hits that they decide to elaborate on.

As requested, we have now performed pathway analysis using STRING to assess functional protein networks and added these findings to Figure 3 —figure supplement 4 and 5. We find highly connected regions in the network corresponding to condensin and histone modifiers in fibroblast hits altering nuclear shape. In contrast, MCF10AT hits showed increased connectivity with nucleoporin proteins. Fibroblast hits displaying an increase in nuclear size identified multiple nucleoporins and MCF10AT hit analysis identified components of DNA replication. We have also expanded our description of some of the hits, especially the ones we elaborate on in the study. We describe these results on p. 11.

4. The need to temper claims about distinct sensitivities of cell types to perturbations that alter nuclear appearance or to provide additional experimental support: The conclusions made from the comparison of the hits from the two independent screens without further testing were found to be inadequately supported. The authors must test a subset of the hits from one cell line in the other (and vice versa) in order to demonstrate that there are indeed cell type specific perturbations. The authors should also compare their list to other published works.

As requested, we now performed side by side experiments in different cell lines to directly compare a subset of nuclear morphology hits in parallel and present the data in Supplemental Figure 3 —figure supplement 1. We find a number of hits that display strong nuclear shape abnormalities in either fibroblasts or MCF10AT cells, but not both with the exception of LMNA, which confirms our screen data. In addition, we compared the hits from our screen with previously published results finding other factors which regulate nuclear morphology to further strengthen our findings. We mention these results on p. 10-11. Despite these results, which confirm our initial claim of cell type differences, we have now toned down these conclusions considering that we have only analyzed two cell lines.

5. Interpretation of the biochemistry: There were several questions about the interpretation of the GST-lamin A biochemistry that need to be addressed. First, it is well established that GST imposes dimerization on proteins expressed as GST fusions independent of cysteines – the findings need to be reconsidered in this light, particularly as it relates to the combined lamin A/lamin C experiments. Second, even in the case that disulfide bonds between cysteines are occurring, there needs to be further consideration of whether this is possibly relevant in vivo (in the reducing environment of the nucleus) and/or requires further characterization, specifically evidence that it is indeed disulfide bonds between the lamin A and not the GST. Third, additional quantitative context for the binding of lamin A to the modified histone tails is needed (are the interactions saturable, what is the realm of the Kd) – as only relative values are provided in Figure 5, the reader currently lacks a framework for knowing what the order of magnitude is for these interactions.

We have now directly addressed the possibility that GST imposes dimerization regardless of the presence of disulfide bonds between cysteine residues in lamin A. To do so, we mutated GST-lamin A cysteine residues to alanine and repeated the histone binding experiments. If the observed binding were artifactually due to GST-mediated dimerization, we should not expect an effect of the cystine mutants on histone binding. We find, however, that the C522A mutation in lamin A results in increased binding of H3 in the presence of lamin C, demonstrating that the observed effects are not due to GST dimerization. We include these data in Figure 4 —figure supplement 1 and discuss the results on p. 13.

We entirely agree with the referee that it is exceptionally challenging to determine the in-vivo relevance of disulfide bonds, not knowing the precise environment of the nucleus. Furthermore, since we were unable to obtain or generate dually-methylated H3-tail peptides, we are not in a position to perform quantitative binding assays between lamin A and the dual-methylation histone H3 tail. Given these concerns, we have now toned down this point on p.14 and discuss the limitations of these findings on p. 19.

As requested, we now include quantitative data in a new Supplementary File 3E for the binding of lamin A to the modified peptides in Figure 5. Note that this assay is not sufficiently quantitative to generate Kds.

6. Need for more depth in the analysis of histone H3.3 variant effects (Figure 6): As these mutants are expressed over WT histones the authors should address how expression level impacts the measured effects on nuclear appearance. What might the ratio of H3.3-V5 to H3.3 be in these experiments (this should be addressable due to the increases size of the V5 on a small histone). In single cells based on the immunostaining, the authors can also address whether there are trends according to the expression level. In this analysis, please also clarify if "shape" is independent of a change in area. There is also a need for statistical analysis. Last, please provide further evidence that the H3 effects manifest through lamins – (also relates to former figures) through localization, relationships to genomic data, etc.

As requested, we have now measured H3.3 mutant levels relative to the total H3.3 levels in the population by immunostaining and single cell analysis. We present these data in Figure 6 —figure supplement 3. By single cell imaging, we find that the mutant levels only slightly increase total H3.3 (Figure 6 —figure supplement 3). We also quantitated, as requested, H3.3-V5 wild-type and mutant levels and compared these parameters to lamin A/C levels and to nuclear shape at the single cell level. We find a slight increase in lamin A/C expression upon expression of both WT and H3.3 mutants and we find no strong correlation between the level of H3.3 mutant expression and effect on shape and size. These data are now described in the text on p. 15 and are included in Figure 6 —figure supplement 3 and 4. We now also indicate that in these experiments shape is independent of change in area and we include statistical analysis of results as requested in Figure 6 —figure supplement 5.

Reviewer #1 (Recommendations for the authors):1. In the introduction of evidence tying nuclear size to nuclear transport additional studies should be referenced in particular the effects of disrupting nuclear export with the small molecule leptomycin B (validated, in fact, by its target XPO1 in the size screen).

As requested, we have included numerous references and discuss on p. 4 the evidence tying nuclear size to nuclear import and export.

2. Please clarify if the lamin A fragment includes the C-terminus after processing or not.

As requested, we now indicate in Supplementary File 1B that the lamin A fragment is the processed mature lamin A construct.

3. The introduction to experiments included in Figure 5 needs to state what form of lamin is being interrogated.

As requested, we now state on p. 14 and 23 the lamin form used is recombinant GST-lamin A (a.a. 506-646).

4. The authors must cite explicit examples with references when suggesting that the presence of hits including nucleoporins and nuclear membrane proteins "validates their screening method".

As requested, we now include on p. 16 several references of nucleoporins and nuclear membrane proteins previously found in similar screens.

Reviewer #2 (Recommendations for the authors):This work is a fantastic advance in the field that provides many novel and deeply interesting findings. However, the paper's large and meaningful amount of data that is completely confused using Z scores (not raw data) and how to deal with variation (no error bars). These two major criticisms of the paper should be addressed before the paper is published.1. There needs to be a clear explanation of the z-score and how it is produced from data for nuclear roundness/shape and size. Overall, the paper would benefit greatly from graphing roundness and size data next to z score to allow the reader to see how the direct measurements (area and perimeter) translate into z score, the measure of change from the mean.

As requested, we now explain in a new Figure 1 —figure supplement 2 in detail how Z-scores were derived from the raw data. In addition, we further describe and clarify statistical methods in the Materials and methods p.22 and in Results section on p.8 and 9. We have now also added Z-scores directly to Figures 1. Furthermore, Figure 1 —figure supplement 4 was added to give a sense of the relationship of phenotypic measurements of area, perimeter, and circularity in comparison to Z-scores generated from the screen. In addition, a new Supplementary Supplementary File 1D with the Z-score next to the raw mean per well circularity score is now included for comparison and mentioned on page 7.

We now explicitly explain that the Z-score is a standard statistical measure used in screening datasets and describes how different an individual sample in the screen is relative to the mean of all samples in the screen PMID: 16869968. For a publicly available standard definition see here. Additional details are provided in the documentation of the R package , used to calculate all the statistics for the screens in this manuscript. We calculated separate Z-scores for either shape or size for each well in the screen using the geometrical measurements as described in the Materials and methods and Results.

More specifically, to obtain a roundness/shape parameter, the circularity of each imaged nucleus was measured using the 4pArea/perimeter^2^ formula. To obtain a size parameter for the nucleus the area of each imaged nucleus was first measured. Raw per nucleus values for all measured parameters were then averaged on a per well (oligo siRNA) basis. Mean per well values were then normalized on a per plate basis. Z-scores for two biological replicates were averaged, thus providing one Z-score values per oligo siRNA. Finally, since each gene is targeted by multiple siRNA oligos in the library, we picked the median of the Z-scores for siRNA oligos against a gene (equivalent to the oligo with 2^nd^ out of 3 strongest biological effect in the assay). This is the Z-score value reported in the manuscript to score hits in the primary screen, and it measures the relative strength of the nuclear morphology change between the knock-out of a particular gene and the median of the population (e.g. Z-score = 0). This approach is standard in the analysis of RNAi screens (see PMID: 16869968). The analysis workflow detailed above is now presented in Figure 1 —figure supplement 2, Figure 1A and is described in detail on p. 22, and 7, 8 and 9.

a. The major problem with this paper lies in how the bulk data is scored and displayed. Z score is not explained anywhere causing major confusion for the reader.

We now describe in a new Figure 1 —figure supplement 2 how the Z-score was generated and have expanded our explanation of the Z-score on pages 7, 8, 9 and 22. Please also see our response to the point above.

b. The core measurement of this paper is area 4 π divided by perimeter squared which is not shown in the paper in any capacity relative to the Z store. It would be gratefully helpful to provide at least one if not a few examples of the type of average or distribution of roundness scores relative to the output Z score. This is important as many previous papers in this field report shape measurements at 1 being a perfect circle and abnormal nuclear morphology measure along this scale from 1 to 0. While abnormal shapes are shown – there connection to the reported Z score and roundness value underlying it is hidden. Same is true for size where exact numbers would be useful to the reader.i. Please show some amount of raw comparative roundness and size measurements.ii. Why not show the > 250 nuclear measurements for scrambled, LMNA, and a few choice hits across the scale of z score?

We now expanded our description of how measurements for nuclear size and shape were made and how Z-scores were derived from the primary measurements. These methods are described in the new Figure 1 —figure supplement 2 and on p. 22.

To further analyze the data from the initial screen, we now provide, as requested, in a new Figure 1 —figure supplement 5 single cell data for nuclear shape hits and controls in histogram format and compare distributions to control cells. We find that the Z-score values are not driven by small subpopulations of cells.

As requested, Z-score values have now been added to figures 1 and Figure 1 —figure supplement 4 to give a sense of the relationship of phenotype and Z-scores. In addition, a new Supplementary File 1D with the Z-score next to the mean per well score was created for comparison.

c. While the first half of the paper reports Z scores (Figure 1-3), later in the paper roundness scores are reported for Figure 6. This is confusing for the reader as measurement of change in nuclear shape and size changes in the paper. This example highlights the importance of being able to bridge the use of these two values, why they are used at different times, and overall understanding. The later histogram like distributions of roundness have no clear statistical measure from the figure.

We apologize for the confusion. As mentioned above, robust Z-scores based on the median of the population of samples are used in analysis of screening data to score treatments/perturbations from large libraries, where most siRNA oligos in the library are expected to have negligible or no biological effect (Figure 1-3). This approach cannot be used in datasets generated by individual or small numbers of experimental conditions that are preselected for biological activity (such as in Figure 6). For clarity, we have added a new Figure 1 —figure supplement 2 explaining how Z-scores are derived from morphological measurements and we are providing a histogram representation of the roundness distribution in Figure 1 —figure supplement 5 and a new Supplementary File 1D with the Z-score next to the mean per well score was created for comparison.

d. THE SPECIFIC question centers around how much the roundness value differs between hits and how much roundness and size vary within one hit. In diagnostic tests of human diseases abnormal nuclear morphology percentages can change by very little between non-aggressive and very aggressive illnesses.i. How much does Z score capture this vs ignore a sub population of abnormally shaped nuclei?

We appreciate these points. To further analyze the data from the initial screen, we now provide, as requested, in a new Figure 1 —figure supplement 5 single cell data for nuclear shape values of several hits and controls in histogram format and compare distributions to control cells. We find that the Z-score values are not driven by small subpopulations of cells.

e. It is appreciated that raw data were made available through GitHub. However, the problem with many raw data sets is that it is hard to find the numbers that matter in a meaningful way.

We have made every effort to provide all relevant data in an easily accessible way. We include GITHUB datasets so readers can evaluate the methods used to generate the data alongside the raw datasets. In addition, we have now added raw data values of the hits identified in the screen in Supplementary File 1D.

2. The manuscript should clearly state why two replicates underscore the majority of the data and that the data graphs provide no error bars.a. The manuscript should justify why two replicates are sufficient for most of the reported data. If you can do two replicates why not have done three to provide the agreed upon triplicate measurement.

It is a standard practice in the field to perform large scale screens such as the ones reported here in duplicate, especially when multiple siRNAs are used in an unpooled fashion (see PMID: 19644458). Our comparison of the first two repeats of the screen indicated very good correspondence between hit lists (see Figure 1 —figure supplement 1). Furthermore, the generated datasets are data-rich and based on several hundred images and analysis of typically 500-1000 cells per sample, generating statistically robust datasets. To further address this point, we have now included in Supplementary File 3A, 3B, and 3C the results of a validation screen using siRNAs with differing target sequences and generated using different chemistry. Additionally, we now build confidence in our hit identification by inclusion of new data in Figure 3 —figure supplement 1 and 2, demonstrating knock-down efficiency of several hits. We typically find reduction levels of 60-90%. We find no relationship between extent of knockdown and nuclear appearance (compare Figure 3 —figure supplement 1 and 2). In summary, screening data was generated by performing the experiment in duplicate, and validation experiments were performed in triplicate such as in Figure 3 —figure supplement 1.

i. The fact that 3 unique siRNAs are used for each KD is not discussed. Does this provide multiple replicates for the paper that are not discussed?

We appreciate this point and have clarified on p. 8 that we indeed targeted each gene with 3 distinct siRNA in an unpooled fashion, thus essentially testing each gene in triplicate and increasing the statistical power of the two replicates of the screen. Hits were defined as showing a statistically significant effect of at least 2 of the 3 siRNAs. Please also see our response to point 2.a above.

b. For z scores, the paper clearly reports two replicates but how are those two replicates are used in the data. The variation from one experiment to the other is not clearly provided. Furthermore, with no triplicate measure should a small population of hits be verified in a more detailed manner like is common in many gene screens? Again, showing just the roundness or size measure would be highly informative.

The variation from one experiment to the other is shown in Figure 1 —figure supplement 1. The Z-scores for each siRNA oligos are calculated as the mean of the two biological replicates. The Z-score on a per gene basis, which is what is reported in this manuscript, is the median of the Z-scores for the 3 oligo siRNA targeting that gene. This Z-score on a per gene basis effectively represents the value of the 2^nd^ siRNA with the strongest biological effect and reduces the chances that a positive hit gene is the result of a single siRNA oligo having strong, but off-targeteffects. To address variation from one dataset to another we have added error bars in several figures (Figures 1, 2, and 5) as well as Supplementary figures. We have now also added a validation screen (Supplementary File 3A, 3B, and 3C) which shows good concordance of Z-score in primary and validation screens.

c. For binding values, there are also no error bars. How is the reader supposed to interpret the data with no idea of variance?

We have added error bars for the binding data.

d. Figure 6 C and D is written in a cryptic manner. Does the manuscript mean to say that compared to wild-type H3.3 all mutant forms (list them) show statistically significant changes in shape C and size D? or are there differences between different mutants? It is unclear.

We have now expanded the description of Figure 6 on p. 14 and 15. We find the H3K9M, H3K27M, and H3K36M mutants alter both nuclear size and shape. While we see subtle differences in their effects, we prefer not to highlight these as they required further investigation.

Reviewer #3 (Recommendations for the authors):1. The screening of Figures1-3 is fine, but I found it important that they restricted analyses "To eliminate hits due to cell death or altered cell-cycle behavior, we excluded any hits with a cell number z-score of less than -2." Some mention of this in the abstract seems important. For example: 'The most dramatic effects were found for cell cycle and death regulators, but we chose to focus on regulators of more subtle effects.' In this regard, the authors need to take great care that the targets identified have zero impact on cell cycle and death.

We agree this is very important and we now mention this caveat in the abstract as suggested by the referee. We found that some hits in our initial shape screen were known regulators of the cell cycle and of mitosis such as *AURKB, MAD2L1* and *CDC2* and displayed a low cell count per well compared to control cells. To eliminate potential false positives, we considered the resulting nuclear morphology changes as secondary effects due to cellular stress, abnormal cell division, siRNA toxicity, and potential lethality as they were often accompanied by significant reduction of cell number. Therefore, we removed nuclear shape and size hits from further analysis if their cell number per well Z-score was below -2. Nuclear shape hits with Z-score below -1.5, described in Figure 1, and no significant changes in cell number, were included in our hit list. We now describe in more detail these criteria and our strategy to reduce false positives and artifacts in the Materials and methods on p. 22 and in the Results sections on p.8 and p. 9.

2. The authors add "reducing agent DTT to inhibit dimerization" of GST-lamin A, but they need to tell us the specific Cys-Cys crossbridges that cause dimerization because GST also has Cys and could be contributing to non-physiological oligomers.

We have now directly addressed the possibility that GST imposes dimerization regardless of the presence of disulfide bonds between cysteine residues. We mutated GST-lamin A cysteine residues to alanine and repeated the histone binding experiments. If the observed binding were artifactually due to GST-mediated dimerization, we should not expect an effect of the cystine mutants on histone binding. We find, however, that the C522A mutation in lamin A results in increased binding of H3 in the presence of lamin C, demonstrating that the observed effects are not due to GST dimerization. We include these data in Figure 4 —figure supplement 1 and discuss the results on p. 13.

3. In analyzing laminA-histone interactions, the authors write "For example, GST-laminA bound ~ 5-fold more efficiently to histone H3R8me2s/K9me2 than histone H3K9me2." The word "efficiently" does not seem a standard descriptor for binding, compared to terms like affinity, specificity, saturation, etc. The authors should strive to show at least one of the 'stronger' interactions is saturable and has a reasonable Kd as a basis for specificity.

This is a good point. Unfortunately, to our knowledge there is no current ChIP-seq human genome map of di-methyl modifications on histone tails. In addit­­­ion, we were unable to generate or procure the individual dually methylated peptides and methyl-methyl H3 antibodies are not available and we are thus not able to perform quantitative binding assays. However, to begin to address this issue, we now provide in a new Supplementary File 3E quantitative data of binding intensities. Note that this assay is not sufficiently quantitative to generate Kds. Given these limitations, we have now toned the claims regarding the methyl-binding sites.

4. The histone-H3 mutation effects on nuclear morphology in Figure 6 are important, but some key details and insights are needed. These seem to be Lenti's that transduced a fraction of the fibroblast cultures, giving levels that are zero, low, or very high in each culture, but does that variation have any effect? Histone levels do relate to cell cycle (e.g. PMID: 35760914), and so are high expressers at later stages of cell cycle (e.g. higher DNA staining) and would that be sensible for how the construct is regulated in its expression? Does the 'shape' parameter neglect differences in 'area'. Are the histone intensities uniform, and what happens to LaminA levels or localization?

We appreciate these points. We generally do not see cell cycle effects in these cells. However, to address some of these issues, we have now measured H3.3 mutant levels relative to the total H3.3 levels in the population by immunostaining and single cell analysis. By single cell imaging, we find that the mutant levels represent a wide range although total H3.3 levels remain relatively consistent. We also quantitated as requested H3.3-V5 wild-type and mutant levels and compared these parameters to lamin A/C levels and to nuclear shape at the single cell level. While we find a slight increase in lamin A/C expression upon expression of both WT and H3.3 mutants, we see no significant effect on lamin A localization. We also see no strong correlation between the level of H3.3 mutant expression and effect on shape and size. These data are now described in the text on p. 14 and p. 15 and are included in Figure 6 —figure supplement 3 and 4. We now also indicate that in these experiments shape is independent of change in area and we include statistical analysis of results as requested in Figure 6 —figure supplement 5.